# Coordination Mechanism of E-Closed-Loop Supply Chain under Social Preference

**Yanhong Qin** * , **Shaojie Wang and Neng Gao**

School of Management, Chongqing University of Technology, Chongqing 400054, China
* Correspondence: qinyanhong24@163.com; Tel.: +86-13594001479

**Abstract:** This study investigates the effect of social preference on the recycling decision and co-ordination in an E-closed-loop supply chain (E-CLSC). Firstly, we set the dynamic E-CLSC game model including a manufacturer, an E-commerce platform (E-platform) and a recycler, where the manufacturer dominates the supply chain and both the E-platform and the recycler are followers. Secondly, we propose an E-CLSC information structure to depict four symmetry and asymmetry cases about the E-platform's fairness concern and the manufacturer's altruistic reciprocity, and the backward induction method is adopted to solve the equilibrium in each case. By comparative analysis, we propose a revenue-sharing and cost-sharing contract to optimize recycling decisions, coordinate the E-CLSC and Pareto-improve all parties' profits. We show that whether information is symmetrical or not, only the wholesale price contract cannot coordinate the E-CLSC, whereas the revenue-sharing and cost-sharing contract can always achieve optimal recycling decisions, coordinate the supply chain and Pareto-improve all parties' profits with a constant cost sharing ratio. In addition, the E-platform's fairness concern can widen the range of the revenue sharing ratio and make it easier to coordinate the E-CLSC, but the manufacturer's altruistic reciprocity may narrow the range of the revenue sharing ratio and make it harder to coordinate the E-CLSC.

**Keywords:** E-closed-loop supply chain; fairness concern; altruistic reciprocity; coordination mechanism



## 1. Introduction

Rapid economic development has accelerated the upgrading of electrical and electronic products, resulting in a rising quantity of discarded electrical and electronic products and causing serious problems such as resource waste and environmental pollution, etc. According to the National Institute of Home Appliances in China, the stock of discarded electrical appliances and electronics in China exceeded 687 million in 2020, and most of these discarded electrical appliances and electronic products were not recycled, resulting in about 800 million tons of solid waste annually. The closed-loop supply chain is an effective way to solve these problems by implementing reverse recycling and remanufacturing. According to the 2021 China Industry Information Report, by the end of 2020, the recycled weight of China's waste electrical and electronic products was 3.8 million tons, with a year-on-year growth of 1.74%, and the recycling value of waste electrical and electronic products was CNY 13.3 billion, with a year-on-year increase of 6.31%. With the rapid development of Internet information technology and e-commerce, many problems have appeared in the network environment of the traditional offline closed-loop supply chain, such as poor information transmission, low consistency, complex links, low efficiency and high cost [1]. For example, the information about recycling mobile phones in each offline store is incomplete, and there is a lack of a unified valuation standard. In the process of mobile phone valuation, problems such as "arbitrary price reduction" and "rough disassemble" are likely to occur, which hinders good operation of the closed-loop supply chain and is not conducive to the recycling and reusing of waste products. An E-closed-loop supply chain (shorten as E-CLSC) can effectively reduce recycling and remanufacturing

costs and improve waste recycling efficiency and economic benefits, thereby effectively enhancing the environmental benefits and promoting sustainable social development by organically combining e-commerce and closed-loop supply chains [2]. More and more enterprises have launched online recycling activities for renewable resources. For example, Huawei relies on its own online shopping mall and third-party recycling platforms to carry out recycling and trading in the old for the new, and Baidu allied with Aihuishou to launch the "Baidu Recycle" platform to recycle various electronic products together [3]. By doing this, they are committing to improving product recovery and reducing surface environmental pollution.

As a key component of overall resource conservation and recovery, the E-CLSC concept has become the focus of social attention. On the one hand, improving the recycling rate is an important goal of E-CLSCs, and supply chain coordination directly determines the stability of E-CLSC operation [4]. On the other hand, social preference as an important psychological factor always significantly affects supply chain decision-making, and fairness concern and altruistic reciprocity are two important and typical types of social preference [5]. An important reason for low recovery efficiency is that some E-CLSC members feel that there is unfair profit distribution and make the decision to reduce the recovery effort, thus reducing the recycling rate and the stability of E-CLSC operation. Specifically, as a key member of the E-CLSC, the e-commerce platform (shortened as E-platform) caring about the fair distribution of channel profits will directly affect the recycling of waste products and the selling of remanufactured products—e.g., as an online recycling platform, "Re life" has no choice but to stop operation due to the unfair distribution of channel profits [6]. Furthermore, dominant manufacturers often implement altruistic and reciprocal behaviors to encourage the E-platform to improve service levels and the recycler to improve recycling rates—e.g., the Tophatter platform implements a minimum price system for settled businesses [7]. Thus, it is necessary to study the recycling decisions and contract coordination of E-CLSCs by considering social preferences and to conduct research more in line with the actual decision-making psychology so as to provide a new analytical perspective for improving recycling rates and promoting E-CLSC operation and waste resource recycling.

This paper sets a three-party E-CLSC model consisting of a manufacturer, an E-platform and a recycler, wherein the manufacturer dominates the supply chain and both the E-platform and the recycler make decisions simultaneously as followers. Then, we propose the E-CLSC information structure to depict four symmetry and asymmetry cases of social preference, i.e., the E-platform's fairness concern and the manufacturer's altruistic reciprocity in the E-CLSC, and the backward induction method is adopted to solve the equilibrium in each case. By comparative analysis, we investigate the effect of fairness concern and altruistic reciprocity on the recycling decision and supply chain coordination under a wholesale price contract. Finally, we propose a revenue-sharing and cost-sharing contract to optimize recycling decisions, coordinate the E-CLSC and Pareto-improve the profits of each member of the E-CLSC.

The remainder of the paper is organized as follows. Section 2 provides a review of the existing literature. Section 3 illustrates our basic model. Section 4 explains the information structure of fairness concern and altruistic reciprocity in the E-CLSC and analyzes the effect of fairness concern and altruistic reciprocity on the recycling decision and supply chain coordination. Section 5 includes the design of the revenue-sharing and cost-sharing contract to achieve optimal recycling decision and supply chain coordination. Section 6 provides a numerical analysis to verify our research conclusions. Section 7 concludes our findings and includes a discussion on the future research direction.

## 2. Literature Review

Although E-CLSCs are widely relevant in reality, most of the current literature still focuses on the recycling decisions and contract coordination of traditional closed-loop supply chains without Internet technology (e.g., E-platforms, big data, cloud computing,

etc.). Furthermore, the application of social preferences in the supply chain has just started, but there are few such applications in closed-loop supply chains, so we will review the literature from two aspects: the recycling decisions and contract coordination of closed-loop supply chains and closed-loop supply chains under social preferences.

(1) The recycling decisions and contract coordination of closed-loop supply chains. Abbey and Blackburn (2015) studied the pricing decision model of a closed-loop supply chain under market segmentation and proved that the entry of reproduction into the market can promote the optimal price rise of new products [8]. Gao et al. (2016) proved that the optimal pricing decision of a closed-loop supply chain is different under different market leading force structures when the market demand is jointly affected by both recovery effort and sales effort [9]. Wang et al. (2019) analyzed the impact of retailers' recycling effort on the closed-loop supply chain pricing decision and illustrated that government incentive can optimize recycling decisions [10]. Gao et al. (2018) proved that improving consumers' recognition of remanufactured products can effectively promote the recycling of waste products, but it is not conducive to improving the overall profit of the supply chain [11]. Zou et al. (2018) studied the recycling decision and contract coordination of a closed-loop supply chain under a carbon trading mechanism [12]. He et al. (2019) studied a closed-loop supply chain under recycling competition and proved that the retailer's competitive behavior has no effect on waste recycling efficiency [4]. Yao and Teng (2019) pointed out that the increase in manufacturers' competition intensity could improve product sales and promote the recycling of waste products in the closed-loop supply chain of two competing manufacturers [13]. Chen et al. (2020) studied the pricing strategy of supply chain members under different cooperation modes [14]. Jia (2020) proved that a government subsidy is conducive to improving the repurchase price and recycling rate of waste products [15]. Xu et al. (2021) proved that increased uncertainty of the remanufacturing output rate is not conducive to the recycling of waste products and found that the revenue-sharing contract based on the Shapley value can achieve closed-loop supply chain coordination [16]. You et al. (2021) studied a three-tier closed-loop supply chain under capital constraints and proved that different risk attitudes of members have different effects on the recovery decision and contract coordination of the closed-loop supply chain [17]. Although the E-CLSC has attracted much attention from the business community and academia, there are few studies related to E-CLSCs. Tian and Yang (2020) analyzed the impact of different power structures and different E-CLSC operation modes on the recycling decisions and cooperation of all parties and found that the recycling efficiency of waste products was the highest when the seller dominated the E-CLSC [18]. Liu (2019) proved that optimizing the cooperation mechanism between manufacturers and platforms can solve the problem of low profits for both sides [1]. Li et al. (2020) designed a price profit sharing coordination mechanism to effectively improve the profits of both a manufacturer and an E-platform so as to ensure the efficient operation of the E-CLSC [3]. Wang et al. (2022) studied E-CLSC recycling decisions under the financial constraint of the E-platform, and they proved that the financial constraint of the E-platform would have a "win–win" effect on the E-platform and the recycler and proposed strategies and suggestions conducive to promoting the recycling of waste products with a case study [19].

(2) The coordination of closed-loop supply chains under social preferences. Many behavioral economic studies have proved that decision-makers are usually affected by social preference. Social preference is the general term for psychological preferences such as fairness concern, altruistic reciprocity, sympathy, envy and pride, of which fairness concern and altruistic reciprocity are two important types. Fairness concern refers to the decision-maker paying attention not only to his own profit but also to the profit comparison with the cared decision-maker. When his own profit is lower than that of the cared decision-maker, it is easy to cause negative unfair utility, so a decision-maker with fairness concerns will make a decision to maximize the total utility including his own direct profit and negative unfair utility. Meanwhile, altruistic reciprocity means that the decision-maker not only pays attention to his own profit but will also try to improve the profit of the cared

decision-maker, so a decision-maker with altruistic reciprocity will make a decision to maximize the total utility including his own profit and positive altruistic utility. There are many examples reflecting the influence of social preference on supply chain decision-making in reality. In 2016, nine airlines announced they would discontinue cooperation with Qunar.com (accessed on 5 January 2016) because the prices and service provided by Qunar.com were unreasonable, which led to unfair psychology of the partners, affected their direct profit and damaged the balance of the overall supply chain operation [20]. Suning E-shop Group tried to use its dominant role to lower the suppliers' wholesale prices so as to increase its own profit margin and attract more consumers with lower sale prices, which caused a serious imbalance in the profit distribution of the supply chain and great dissatisfaction of suppliers [1]. Suning has since put forward a strategy of "altruistic integration", which includes building a full data chain supporting system, building a broader space for cooperation and encrypting a grassroot logistics network for exploration and innovation from multiple dimensions so as to build a complete "altruistic" logistics system so that different market players can "hitchhike" to achieve mutual benefits and win–win results.

In terms of fairness concern, Yao et al. (2020) proved that a manufacturer's fairness concern is not conducive to recycling waste products [21]. Wang et al. (2019) proved that the increased recycler's fairness concern would reduce the recycling rate of waste products [22], and similar conclusions were obtained in some research such as [23,24], etc. Zheng et al. (2019a) and Zheng (2019b) proved that if the acceptance rate of remanufactured products is low, the allocation mechanism based on the variable weight Shapley value can achieve supply chain coordination [25,26]. Huang (2020) proved that the retailer's fairness concern has a significant impact on closed-loop supply chain coordination using the differential game theory [27]. Sarkar and Bhala (2021) proved that only when the retailer shows strong enough fairness concern can a closed-loop supply chain be coordinated [28]. Li (2021) proved that as long as the revenue sharing ratio is within a certain range, a revenue-sharing contract can achieve Pareto-improvement of a green closed-loop supply chain under fairness concern [29]. Shu et al. (2020) paid attention to the impact of vertical distribution fairness and horizontal induction fairness on the profit distribution of a closed-loop supply chain at the same time and proved that the recycler can obtain more profit with vertical fairness concern but may make the supply chain coordination deviate from the optimal status [30]. Only little research has referred to E-CLSC. Wang et al. (2019) proved that an E-platform's fairness concern can obtain more profit while reducing the recycling rate of waste products in an E-CLSC dominated by the E-platform [10]. Furthermore, Wang et al. (2021) analyzed the impact of fairness concern and consumers' low carbon awareness on E-CLSC recycling decisions [20].

In terms of altruistic reciprocity, Zhang (2015) studied the effect of altruistic reciprocity on the pricing and channel efficiency of a closed-loop supply chain and illustrated that altruistic reciprocity could improve the channel operation efficiency of the supply chain [31]. Li et al. (2021) set a differential game model of a closed-loop supply chain under altruistic reciprocity according to the dynamic recycling rate, and they proved that the manufacturer's altruistic reciprocity has a positive impact on the supply chain profit of different recycling modes and the recovery rate of waste products [32]. Su et al. (2020) analyzed the impact of altruistic reciprocity on the incentive mechanism of the construction industry based on the principal–agent framework and proved that altruistic reciprocity can alleviate the problem of asymmetrical information by saving the remanufacturer's costs and improving the recycler's recycling efficiency [33]. Ding et al. (2022) studied the impact of altruistic reciprocity on the recovery efficiency and system profit distribution of a closed-loop supply chain based on scale diseconomies and proved that whether recovery efficiency can be improved depends on the supplier's altruistic reciprocity intensity and the manufacturer's attitude towards the supplier's altruistic reciprocity [34]. Similar to fairness concern, there are only few studies related to altruistic reciprocity in E-CLSCs. Zhang et al. (2019) set an E-CLSC recycling model by considering altruistic reciprocity based on different power

structures, and they proved that altruistic reciprocity has a dual role in increasing the profit of each member in the E-CLSC [35]. Lan and Zheng (2020) introduced altruistic reciprocity into an E-CLSC and proposed a coordination mechanism based on a cooperative game to achieve a fairer distribution of residual profit among supply chain members [36]. Wang et al. (2022) designed a contract of "quantity discount combined with fixed cost sharing" to achieve E-CLSC coordination by considering the effect of the dominant manufacturer's altruistic reciprocity on recycling decision under a government incentive mechanism [20].

There are many studies on the traditional closed-loop supply chain, but only few about the E-CLSC. There exists a significant difference between a traditional closed-loop supply chain and an E-CLSC, such as the operation structure, cooperation mode, recycling process, etc. Therefore, it is difficult to apply the conclusions of traditional closed-loop supply chains to E-CLSCs directly. With the development of network and information technology, it is necessary to optimize recycling decisions and supply chain coordination specially for E-CLSCs in line with the development of e-commerce and big data so as to effectively reduce recycling and remanufacturing costs, improve waste recovery efficiency and economic benefits for enterprises and thus effectively improve sustainable social development and enhance the environmental benefits. Additionally, the existing research does not involve the improvement of recycling rates and supply chain coordination at the same time, while improving the recycling rate is an important goal of E-CLSCs, and supply chain coordination directly determines the stability of the E-CLSC operation.

Furthermore, some research does refer to the effect of fairness concern or altruistic reciprocity on recycling decision and contract coordination, but there are two problems: firstly, there are too few studies introducing fairness concern or altruistic reciprocity into E-CLSCs (e.g., [10,20] on fairness concern and [19,35,36] on altruistic reciprocity), and these studies do not cover the impact of fairness concern and altruistic reciprocity on recycling decision and contract coordination at the same time. Secondly, these studies assume that the social preference information is symmetrical, but social preference comprises subjective and private information, and there are problems of deliberate concealment and disguise. Therefore, it is necessary to investigate the effect of fairness concern and altruistic reciprocity on the recycling decision and contract coordination of E-CLSCs simultaneously under symmetrical and asymmetrical information.

Our contribution lies in four aspects as below:

Firstly, we set an E-CLSC model including a manufacturer, an E-platform and a recycler and investigate recycling decisions and contract coordination to meet the development of e-commerce and big data technology, optimize recycling decisions and promote sustainable E-CLSC operation.

Secondly, we investigate the impact of two typical social preferences (i.e., fairness concern and altruistic reciprocity) on the recycling decision and contract coordination in the E-CLSC, which is more in line with actual decision-making psychology and can thus provide a new analytical perspective for improving the recycling rate, promoting E-CLSC operation and recovering resources.

Thirdly, we analyze the impact of social preferences as subjective private information on recycling decision and contract coordination in the E-CLSC by depicting four cases of symmetrical and asymmetrical information under fairness concern and altruistic reciprocity with information structure and prove that whether social preference information is symmetrical or not, the wholesale price contract cannot coordinate the E-CLSC, but once the decision-maker can consider the partner's social preference to make decisions, it is conducive to optimizing the decision of all parties and the supply chain.

Finally, we design a revenue-sharing and cost-sharing contract to achieve the optimal recycling decision, supply chain coordination and Pareto-improvement of each member's profit regardless of whether the information is symmetrical or not. We prove that the E-platform's fairness concern will enlarge the range of the revenue sharing proportion and reduce the coordination difficulty of the E-CLSC, while the manufacturer's altruis-

tic reciprocity will shrink the range of the revenue sharing proportion and increase the coordination difficulty of the E-CLSC.

## 3. Basic Model

### 3.1. Model Assumption and Denotation

The E-CLSC consists of a manufacturer, an E-platform and a recycler, as shown in Figure 1a. The manufacturer produces new products and remanufactured products and sells them to the E-platform simultaneously at a certain wholesale price. The E-platform (e.g., JD Mall, Xiaomi Mall) sells new products and remanufactured products through an online channel while providing consumers with services such as mailing, online trading, after-sale support, information consulting and recycling advertisement. The recycler is responsible for recycling waste products and then delivering them to the manufacturer at a certain transfer price, as shown in Figure 1a. The manufacturer dominates the E-CLSC and is in the leading position; both the E-platform and the recycler are in the following position at the same time, and they form a Stackelberg game. The game sequence is as follows: the manufacturer decides the wholesale price $w$ in advance; then, the E-platform decides the retail price $p$ and service level $s$, and the recycler decides the recycling rate $t$ of waste products (as shown in Figure 1b).

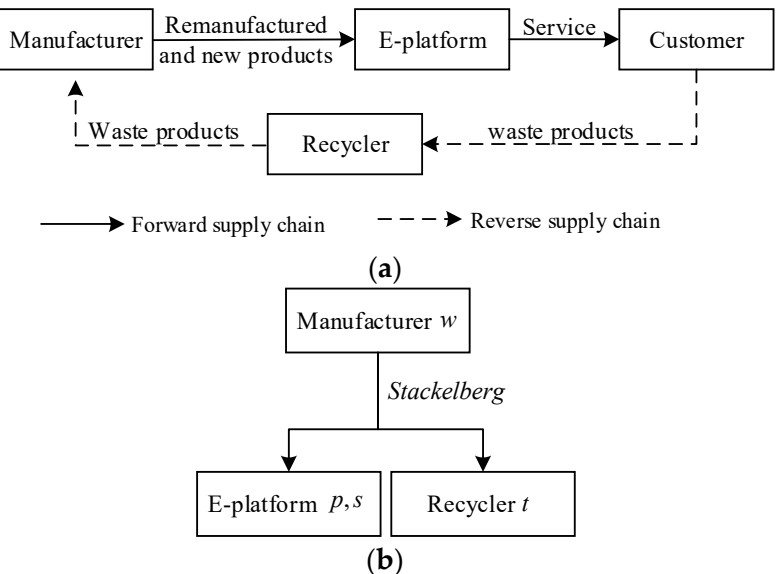

**Figure 1.** (**a**) The operation mode of the E-CLSC. (**b**) Game sequence of the E-CLSC.

The model assumptions are as follows:

(1) The recycler always tests the recycled waste products strictly to ensure that all the recycled waste products can be renovated and processed into remanufactured products and achieve no difference with the new products in appearance and function and can be sold to consumers at the same price [3,7]. Although the recycled products cannot be recovered to perfectly new standard through remanufacturing, they can meet the requirements for normal use of consumers after special remanufacture. If we consider the difference between new products and reprocessed products, we need to add another relevant parameter, namely the proximity to the new products or the substitution degree with the new products, which will increase the model complexity seriously; at the same time, we cannot significantly analyze the impact of social preferences on the recycling decision. Therefore, we assume that the recycled products have no difference with the new products through remanufacturing, which reduces the complexity of the model and makes the effect of social preference on recycling decision more obvious and direct.

(2) The unit manufacturing cost of new products is $c_m$, and the unit manufacturing cost of remanufactured products is $c_r$, which means that the unit remanufacturing cost

$\Delta = c_m - c_r$ can be saved. The recycler collects waste products from consumers at the recycling price $\psi$ and then transfers the waste products to the manufacturer at the transfer price $b$. Thus, the manufacturer's unit manufacturing cost is $c = c_m - t(\Delta - b)$. For simplification, $b$ and $\psi$ are exogenous and determined by the market in advance. To make remanufacturing meaningful, we assume that $\Delta > b > \psi > 0$.

(3) The market demand faced by the E-platform is $q = \alpha - p + \gamma s$, where $\alpha(\alpha > p)$ denotes the maximal potential market demand and $\gamma$ is the service-effect coefficient [20]. Generally, consumers are more sensitive to the retail price, so $0 < \gamma < 1$.

(4) The E-platform will take the service cost $c(s) = \frac{1}{2}s^2$ as it provides various information and sales services, and the recycler will spend the cost $c(t) = \frac{1}{2}t^2$ as it provides recycling services such as collecting recycling information, releasing recycling facilities, etc. [19,20].

(5) To ensure that all members participate in the game, the parameters should meet the constraint $1 - \gamma^2 - 2(\Delta - \psi)^2 > 0$.

(6) $\pi_m$, $\pi_e$, $\pi_r$ and $\pi_{sc}$ denote the profits of the manufacturer, the E-platform, the recycler and the E-CLSC, respectively. $u_m$ and $u_e$ denote the utility of the manufacturer and the E-platform, respectively. Superscript $C$ denotes the centralized decision-making mode, $RC$ denotes the revenue-sharing and cost-sharing contract and $*$ denotes the equilibrium solution. Superscripts $N$ and $F$ denote complete rationality and fairness concern for the E-platform, respectively; $M$ and $L$ denote complete rationality and altruistic reciprocity for the manufacturer, respectively; and $S$ and $A$ denote corresponding symmetrical and asymmetrical information, respectively.

Based on the above descriptions and assumptions, the profits of the manufacturer, the E-platform, the recycler and the E-CLSC can be denoted as follows:

$$\pi_m = [w - c_m + t(\Delta - b)]q \tag{1}$$

$$\pi_e = (p - w)q - \frac{1}{2}s^2 \tag{2}$$

$$\pi_r = (b - \psi)qt - \frac{1}{2}t^2 \tag{3}$$

$$\pi_{sc} = [p - c_m + t(\Delta - \psi)]q - \frac{1}{2}s^2 - \frac{1}{2}t^2 \tag{4}$$

### 3.2. Basic Model

Under the centralized decision-making mode, the manufacturer, E-platform and recycler cooperate as a whole to maximize the E-CLSC's profits to determine the optimal retail price $p$, service level $s$ and waste recycling rate $t$. Therefore, the decision-making problem of the E-CLSC can be denoted as follows:

$$\underset{p,s,t}{Max}\pi_{sc}^C = [p - c_m + t(\Delta - \psi)](\alpha - p + \gamma s) - \frac{1}{2}s^2 - \frac{1}{2}t^2 \tag{5}$$

The Hessian matrix of $\pi_{sc}^C$ is $H_{SC}^C = \begin{bmatrix} -2 & \gamma & -(\Delta - \psi) \\ \gamma & -1 & \gamma(\Delta - \psi) \\ -(\Delta - \psi) & \gamma(\Delta - \psi) & -1 \end{bmatrix}$, for $H_{SC1}^C = -2 < 0$, $H_{SC2}^C = 2 - \gamma^2 > 0$ and $H_{SC3}^C = -[2 - \gamma^2 - (\Delta - \psi)^2] < 0$; thus, $\pi_{sc}^C$ is strictly concave in $p$, $s$ and $t$. Let $\frac{\partial \pi_{sc}^C}{\partial p} = 0$, $\frac{\partial \pi_{sc}^C}{\partial s} = 0$ and $\frac{\partial \pi_{sc}^C}{\partial t} = 0$; we can compute the unique and optimal solutions $p^{C*}$, $s^{C*}$ and $t^{C*}$ as follows: $p^{C*} = \frac{\alpha[1 - (\Delta - \psi)^2] + c_m(1 - \gamma^2)}{2 - \gamma^2 - (\Delta - \psi)^2}$, $s^{C*} = \frac{\gamma(\alpha - c_m)}{2 - \gamma^2 - (\Delta - \psi)^2}$, $t^{C*} = \frac{(\alpha - c_m)(\Delta - \psi)}{2 - \gamma^2 - (\Delta - \psi)^2}$ and $\pi_{sc}^{C*} = \frac{(\alpha - c_m)^2}{2[2 - \gamma^2 - (\Delta - \psi)^2]}$.

Under the decentralized decision-making mode without social preference, the E-CLSC game model is denoted as

$$\underset{w}{Max}\pi_m = [w - c_m + t(\Delta - b)](\alpha - p + \gamma s) \tag{6}$$

$$s.t \begin{cases} \underset{p,s}{Max}\pi_e = (p - w)(\alpha - p + \gamma s) - \frac{1}{2}s^2 \\ \underset{t}{Max}\pi_r = (b - \psi)(\alpha - p + \gamma s)t - \frac{1}{2}t^2 \end{cases} \tag{7}$$

The backward induction method is used to solve the problem. In the second stage of the game, the optimal response function of the E-platform and the recycler is

$$p^*(w) = \frac{\alpha + (1 - \gamma^2)w}{2 - \gamma^2}, \; s^*(w) = \frac{\gamma(\alpha - w)}{2 - \gamma^2}, \; t^*(w) = \frac{(b - \psi)(\alpha - w)}{2 - \gamma^2}$$

Taking $p^*(w)$, $s^*(w)$ and $t^*(w)$ into the manufacturer's target function (6), the optimal wholesale price is

$$w^* = \frac{\alpha[2 - \gamma^2 + 2(b - \Delta)(b - \psi)] + c_m(2 - \gamma^2)}{2[2 - \gamma^2 + (b - \Delta)(b - \psi)]} \tag{8}$$

Thus, we can obtain the optimal $p^*$, $s^*$ and $t^*$ in equilibrium as follows:

$$p^* = \frac{\alpha[3 - \gamma^2 + 2(b-\Delta)(b-\psi)] + c_m(1-\gamma^2)}{2[2 - \gamma^2 + (b-\Delta)(b-\psi)]}$$

$$s^* = \frac{\gamma(\alpha - c_m)}{2[2 - \gamma^2 + (b-\Delta)(b-\psi)]}$$

$$t^* = \frac{(\alpha - c_m)(b - \psi)}{2[2 - \gamma^2 + (b-\Delta)(b-\psi)]}$$

$(w^*, p^*, s^*, t^*)$ is the game equilibrium solution; taking $(w^*, p^*, s^*, t^*)$ into the profit functions (1)–(3), we can compute the optimal profit in the E-CLSC as below:

$$\pi_m^* = \frac{(\alpha - c_m)^2}{4[2 - \gamma^2 + (b-\Delta)(b-\psi)]}$$

$$\pi_e^* = \frac{(2 - \gamma^2)(\alpha - c_m)^2}{8[2 - \gamma^2 + (b-\Delta)(b-\psi)]^2}$$

$$\pi_r^* = \frac{(b-\psi)^2(\alpha - c_m)^2}{8[2 - \gamma^2 + (b-\Delta)(b-\psi)]^2}$$

It is easy to prove $\pi_m^* > \pi_e^* > \pi_r^*$. On the one hand, both the E-platform and the recycler are disadvantaged in the E-CLSC profit distribution, so they are likely to generate fairness concern and affect the recycling decision. The E-platform as the key E-CLSC member can affect the positive sales and reverse recycle, and thus, the E-platform is more likely to cause negative unfair utility. On the other hand, the dominant manufacturer would like to implement altruistic reciprocity behavior to improve the profits of both the E-platform and the recycler. In particular, the manufacturer is more likely to implement altruistic reciprocity behavior to help the key member, the E-platform. Therefore, in order to simplify the model, we only consider the E-platform's fairness concern to the manufacturer and the manufacturer's altruistic reciprocity to the E-platform.

According to [10,37], the E-platform's utility function is $u_e = \pi_e - \theta(\pi_m - \pi_e)$, and $\theta(0 \le \theta \le 1)$ is the fairness concern intensity. Simultaneously, the manufacturer's utility function is $u_m = \pi_m + \varphi\pi_e$, and $\varphi(0 \le \varphi \le \frac{1}{2})$ is the altruistic reciprocity intensity [38].

## 4. Model under Information Structure

### 4.1. Fairness Concern Scenario

According to Qin et al. (2021), as fairness concern is internal and subjective psychological information, the manufacturer may not necessarily have the true information about the E-platform's fairness concern intensity. According to the actual situation of whether the E-platform is concerned about fairness and the manufacturer's perception about the E-platform's fairness concern, the information structure of the E-CLSC's fairness concern can be classified as four cases, as noted below.

① Case I: The E-platform is completely rational, and this information is symmetrical. The E-platform's decision problem is $\underset{p,s}{Max}\pi_e^{NS} = \pi_e = (p-w)q - \frac{1}{2}s^2$, which is common knowledge in the E-CLSC. Case I is equivalent to a decentralized decision in the basic model, and the response function is $p^{NS*}(w) = \frac{\alpha+(1-\gamma^2)w}{2-\gamma^2}$, $s^{NS*}(w) = \frac{\gamma(\alpha-w)}{2-\gamma^2}$ and $t^{NS*}(w) = \frac{(b-\psi)(\alpha-w)}{2-\gamma^2}$. Thus, the optimal solution in equilibrium is as follows:

$$w^{NS*} = \frac{\alpha[2-\gamma^2+2(b-\Delta)(b-\psi)]+c_m(2-\gamma^2)}{2[2-\gamma^2+(b-\Delta)(b-\psi)]}$$

$$p^{NS*} = \frac{\alpha[3-\gamma^2+2(b-\Delta)(b-\psi)]+c_m(1-\gamma^2)}{2[2-\gamma^2+(b-\Delta)(b-\psi)]}$$

$$s^{NS*} = \frac{\gamma(\alpha-c_m)}{2[2-\gamma^2+(b-\Delta)(b-\psi)]}$$

$$t^{NS*} = \frac{(\alpha-c_m)(b-\psi)}{2[2-\gamma^2+(b-\Delta)(b-\psi)]}$$

$(w^{NS*}, p^{NS*}, s^{NS*}, t^{NS*})$ is the optimal solution of equilibrium in Case I. We can calculate the profit as follows:

$$\pi_m^{NS*} = \frac{(\alpha-c_m)^2}{4[2-\gamma^2+(b-\Delta)(b-\psi)]}$$

$$\pi_e^{NS*} = \frac{(2-\gamma^2)(\alpha-c_m)^2}{8[2-\gamma^2+(b-\Delta)(b-\psi)]^2}$$

$$\pi_r^{NS*} = \frac{(b-\psi)^2(\alpha-c_m)^2}{8[2-\gamma^2+(b-\Delta)(b-\psi)]^2}$$

$$\pi_{sc}^{NS*} = \frac{(\alpha-c_m)^2[3(2-\gamma^2)+2(b-\Delta)(b-\psi)+(b-\psi)^2]}{8[2-\gamma^2+(b-\Delta)(b-\psi)]^2}$$

② Case II: The E-platform is completely rational, but this information is asymmetric. The manufacturer infers the E-platform's decision problem as $\underset{p,s}{Max}u_e^{NA'} = \pi_e - \theta(\pi_m - \pi_e)$, but the E-platform's real decision problem is $\underset{p,s}{Max}\pi_e^{NA} = \pi_e = (p-w)q - \frac{1}{2}s^2$. Under asymmetrical information, the manufacturer thinks that the E-platform's utility is $u_e^{NA'}$, and thus, the Hessian matrix is $H_e^{NA} = \begin{bmatrix} -2(1+\theta) & \gamma(1+\theta) \\ \gamma(1+\theta) & -(1+\theta) \end{bmatrix}$, for $H_{e1}^{NA} = -2(1+\theta) < 0$ and $H_{e2}^{NA} = (1+\theta)^2(2-\gamma^2) > 0$. $H_e^{NA}$ is a negative matrix and $u_e^{NA'}$ is strictly concave, so the manufacturer thinks the optimal response function of the E-platform and the recycler is as below:

$$p^{NA'*}(w) = \frac{(1+\theta)[(1-\gamma^2)w+\alpha]+\theta(1-\gamma^2)[w-c_m+t(\Delta-b)]}{(1+\theta)(2-\gamma^2)}$$

$$s^{NA'*}(w) = \frac{\gamma[\theta t(b-\Delta)+(1+\theta)(\alpha-w)+\theta(c_m-w)]}{(1+\theta)(2-\gamma^2)}$$

$$t^{NA'*}(w) = \frac{(b-\psi)[(1+\theta)\alpha+\theta c_m-(1+2\theta)w]}{(1+\theta)(2-\gamma^2)-\theta(b-\Delta)(b-\psi)}$$

Take $p^{NA'*}(w)$, $s^{NA'*}(w)$ and $t^{NA'*}(w)$ into $\underset{w}{Max}\,\pi_m = [w - c_m + t^{NA'*}(w)(\Delta - b)](\alpha - p^{NA'*}(w) + \gamma s^{NA'*}(w))$ in the first stage; for $\frac{\partial^2 \pi_m^{NA'}}{\partial w^2} = -\frac{2(1+\theta)(1+2\theta)[2-\gamma^2+(b-\Delta)(b-\psi)]}{[(1+\theta)(2-\gamma^2)-\theta(b-\Delta)(b-\psi)]^2} < 0$, we can obtain the optimal wholesale price as follows:

$$w^{NA*} = \frac{\alpha[(1+\theta)(2-\gamma^2)+(2+3\theta)(b-\Delta)(b-\psi)]+c_m[(1+3\theta)(2-\gamma^2)+\theta(b-\Delta)(b-\psi)]}{2(1+2\theta)[2-\gamma^2+(b-\Delta)(b-\psi)]}$$

In fact, the E-platform makes a decision under $\underset{p,s}{Max}\,\pi_e^{NA} = \pi_e = (p-w)q - \frac{1}{2}s^2$, so the real response function of the E-platform and the recycler is $p^{NA*}(w) = p^{NS*}(w) = \frac{\alpha+(1-\gamma^2)w}{2-\gamma^2}$, $s^{NA*}(w) = s^{NS*}(w) = \frac{\gamma(\alpha-w)}{2-\gamma^2}$ and $t^{NA*}(w) = t^{NS*}(w) = \frac{(b-\psi)(\alpha-w)}{2-\gamma^2}$. Taking $w^{NA*}$ into $p^{NA*}(w)$, $s^{NA*}(w)$ and $t^{NA*}(w)$, we can obtain the optimal solution in equilibrium as follows:

$$w^{NA*} = \frac{\alpha[(1+\theta)(2-\gamma^2)+(2+3\theta)(b-\Delta)(b-\psi)]+c_m[(1+3\theta)(2-\gamma^2)+\theta(b-\Delta)(b-\psi)]}{2(1+2\theta)[2-\gamma^2+(b-\Delta)(b-\psi)]}$$

$$p^{NA*} = \frac{\begin{array}{c}\theta\alpha\gamma^4 + 3\theta\alpha[1 - \gamma^2(b-\Delta)(b-\psi)] + 7\theta\alpha[1 - \gamma^2 + (b-\Delta)(b-\psi)] + \alpha(2-\gamma^2) \\ [3 - \gamma^2 + 2(b-\Delta)(b-\psi)] + c_m(1-\gamma^2)[(1+3\theta)(2-\gamma^2)+\theta(b-\Delta)(b-\psi)]\end{array}}{2(1+2\theta)(2-\gamma^2)[2-\gamma^2+(b-\Delta)(b-\psi)]}$$

$$s^{NA*} = \frac{\gamma(\alpha-c_m)[(1+3\theta)(2-\gamma^2)+\theta(b-\Delta)(b-\psi)]}{2(1+2\theta)(2-\gamma^2)[2-\gamma^2+(b-\Delta)(b-\psi)]}$$

$$t^{NA*} = \frac{(b-\psi)(\alpha-c_m)[(1+3\theta)(2-\gamma^2)+\theta(b-\Delta)(b-\psi)]}{2(1+2\theta)(2-\gamma^2)[2-\gamma^2+(b-\Delta)(b-\psi)]}$$

$(w^{NA*}, p^{NA*}, s^{NA*}, t^{NA*})$ is the optimal solution of equilibrium in Case II. We can calculate the profit as follows:

$$\pi_e^{NA*} = \frac{(\alpha-c_m)^2[(1+3\theta)(2-\gamma^2)+\theta(b-\Delta)(b-\psi)]^2}{8(2-\gamma^2)(1+2\theta)^2[2-\gamma^2+(b-\Delta)(b-\psi)]^2}$$

$$\pi_m^{NA*} = \frac{(\alpha-c_m)^2[(1+3\theta)(2-\gamma^2)+\theta(b-\Delta)(b-\psi)][(1+\theta)(2-\gamma^2)-\theta(b-\Delta)(b-\psi)]}{4[(1+2\theta)(2-\gamma^2)]^2[2-\gamma^2+(b-\Delta)(b-\psi)]}$$

$$\pi_r^{NA*} = \frac{[(\alpha-c_m)(b-\psi)]^2[(1+3\theta)(2-\gamma^2)+\theta(b-\Delta)(b-\psi)]^2}{8[(1+2\theta)(2-\gamma^2)]^2[2-\gamma^2+(b-\Delta)(b-\psi)]^2}$$

$$\pi_{sc}^{NA*} = \frac{\begin{array}{c}(\alpha - c_m)^2[(1 + 3\theta)(2 - \gamma^2) + \theta(b-\Delta)(b - \psi)]\{5\theta(2 - \gamma^2)^2 + \theta(b-\Delta)(b-\psi)[2 - \gamma^2 - \\ (b-\Delta)(b-\psi)] + \theta(b-\psi)^2[3(2-\gamma^2) - (\psi-\Delta)(b-\Delta)] + (2-\gamma^2)[3(2-\gamma^2) + 2(b-\Delta)(b-\psi) + (b-\psi)^2]\}\end{array}}{8[(1+2\theta)(2-\gamma^2)]^2[2-\gamma^2+(b-\Delta)(b-\psi)]^2}$$

③ Case III: The E-platform shows fairness concern, and this information is symmetric. The E-platform's decision is common knowledge in the E-CLSC, as below:

$$\underset{p,s}{Max}\,u_e^{FS} = \pi_e - \theta(\pi_m - \pi_e) = (1+\theta)[(p-w)(\alpha - p + \gamma s) - \frac{1}{2}s^2] - \theta[w - c_m + t(\Delta - b)](\alpha - p + \gamma s)$$

It is easy to compute the response function in the second stage as follows:

$$t^{FS*}(w) = \frac{(b-\psi)[(1+\theta)\alpha+\theta c_m-(1+2\theta)w]}{(1+\theta)(2-\gamma^2)-\theta(b-\Delta)(b-\psi)}$$

$$p^{FS*}(w) = \frac{(1+\theta)[(1-\gamma^2)w+\alpha]+\theta(1-\gamma^2)[w-c_m+t(\Delta-b)]}{(1+\theta)(2-\gamma^2)}$$

$$s^{FS*}(w) = \frac{\gamma[\theta t(b-\Delta)+(1+\theta)(\alpha-w)+\theta(c_m-w)]}{(1+\theta)(2-\gamma^2)}$$

Thus, the optimal solution is as follows, by the backward induction method:

$$w^{FS*} = \frac{\alpha[(1+\theta)(2-\gamma^2)+(2+3\theta)(b-\Delta)(b-\psi)]+c_m[(1+3\theta)(2-\gamma^2)+\theta(b-\Delta)(b-\psi)]}{2(1+2\theta)[2-\gamma^2+(b-\Delta)(b-\psi)]}$$

$$p^{FS*} = \frac{\alpha[3-\gamma^2+2(b-\Delta)(b-\psi)]+c_m(1-\gamma^2)}{2[2-\gamma^2+(b-\Delta)(b-\psi)]}$$

$$s^{FS*} = \frac{\gamma(\alpha-c_m)}{2[2-\gamma^2+(b-\Delta)(b-\psi)]}$$

$$t^{FS*} = \frac{(\alpha-c_m)(b-\psi)}{2[2-\gamma^2+(b-\Delta)(b-\psi)]}$$

$(w^{FS*}, p^{FS*}, s^{FS*}, t^{FS*})$ is the optimal solution of equilibrium in Case III. We can calculate the profit as follows:

$$\pi_m^{FS*} = \frac{(1+\theta)(\alpha-c_m)^2}{4(1+2\theta)[2-\gamma^2+(b-\Delta)(b-\psi)]}$$

$$\pi_e^{FS*} = \frac{(\alpha-c_m)^2[(1+4\theta)(2-\gamma^2)+2\theta(b-\Delta)(b-\psi)]}{8(1+2\theta)[2-\gamma^2+(b-\Delta)(b-\psi)]^2}$$

$$\pi_r^{FS*} = \frac{[(b-\psi)(\alpha-c_m)]^2}{8[2-\gamma^2+(b-\Delta)(b-\psi)]^2}$$

$$\pi_{sc}^{FS*} = \frac{(\alpha-c_m)^2[3(2-\gamma^2)+2(b-\Delta)(b-\psi)+(b-\psi)^2]}{8[2-\gamma^2+(b-\Delta)(b-\psi)]^2}$$

④ Case IV: The E-platform shows fairness concern, but this information is asymmetric. The manufacturer infers the E-platform's decision problem as $\underset{p,s}{Max}\pi_e^{FA'} = \pi_e$, but the E-platform's real decision problem is $\underset{p,s}{Max}u_e^{FA} = \pi_e - \theta(\pi_m - \pi_e)$. The manufacturer infers the E-platform's response function as in Case I—i.e., $p^{FA'*}(w) = p^{NS*}(w)$, $s^{FA'*}(w) = s^{NS*}(w)$ and $t^{FA'*}(w) = t^{NS*}(w)$; thus, the optimal wholesale price is $w^{FA*} = \frac{\alpha[2-\gamma^2+2(b-\Delta)(b-\psi)]+c_m(2-\gamma^2)}{2[2-\gamma^2+(b-\Delta)(b-\psi)]}$. With asymmetrical information, the E-platform's real response function is the same as in Case III—i.e., $p^{FA*}(w) = p^{FS*}(w)$, $s^{FA*}(w) = s^{FS*}(w)$ and $t^{FA*}(w) = t^{FS*}(w)$. Taking $w^{FA*}$ into each response function, the optimal solution in equilibrium is as follows:

$$p^{FA*} = \frac{\begin{aligned}&2\theta[\alpha(2-\gamma^2)^2-\alpha(b-\Delta)^2(b-\psi)^2+(\alpha-c_m)(1-\gamma^2)(b-\Delta)(b-\psi)]\\&+(2-\gamma^2)[\alpha(3-\gamma^2)+2\alpha(b-\Delta)(b-\psi)+c_m(1-\gamma^2)]\end{aligned}}{2[2-\gamma^2+(b-\Delta)(b-\psi)][(1+\theta)(2-\gamma^2)-\theta(b-\Delta)(b-\psi)]}$$

$$s^{FA*} = \frac{\gamma(\alpha-c_m)[2-\gamma^2+2\theta(\Delta-b)(b-\psi)]}{2[2-\gamma^2+(b-\Delta)(b-\psi)][(1+\theta)(2-\gamma^2)-\theta(b-\Delta)(b-\psi)]}$$

$$t^{FA*} = \frac{(\alpha-c_m)(b-\psi)[2-\gamma^2+2\theta(\Delta-b)(b-\psi)]}{2[2-\gamma^2+(b-\Delta)(b-\psi)][(1+\theta)(2-\gamma^2)-\theta(b-\Delta)(b-\psi)]}$$

$(w^{FA*}, p^{FA*}, s^{FA*}, t^{FA*})$ is the optimal solution of equilibrium in Case IV. We can calculate the profit as follows:

$$\pi_m^{FA*} = \frac{(1+\theta)(2-\gamma^2)(\alpha-c_m)^2[2-\gamma^2+2\theta(\Delta-b)(b-\psi)]}{4[2-\gamma^2+(b-\Delta)(b-\psi)][(1+\theta)(2-\gamma^2)-\theta(b-\Delta)(b-\psi)]^2}$$

$$\pi_e^{FA*} = \frac{(1+2\theta)[(2-\gamma^2)(\alpha-c_m)]^2[2-\gamma^2+2\theta(\Delta-b)(b-\psi)]}{8[2-\gamma^2+(b-\Delta)(b-\psi)]^2[(1+\theta)(2-\gamma^2)-\theta(b-\Delta)(b-\psi)]^2}$$

$$\pi_r^{FA*} = \frac{[(\alpha-c_m)(b-\psi)]^2[2-\gamma^2+2\theta(\Delta-b)(b-\psi)]}{8[2-\gamma^2+(b-\Delta)(b-\psi)]^2[(1+\theta)(2-\gamma^2)-\theta(b-\Delta)(b-\psi)]^2}$$

$$\pi_{sc}^{FA*} = \frac{\begin{aligned}&(\alpha-c_m)^2[2-\gamma^2+2\theta(\Delta-b)(b-\psi)]\{2\theta(2-\gamma^2)[2(2-\gamma^2)+(b-\Delta)(b\\&-\psi)]-2\theta(b-\Delta)(b-\psi)^3+(2-\gamma^2)[3(2-\gamma^2)+2(b-\Delta)(b-\psi)+(b-\psi)^2]\}\end{aligned}}{8[2-\gamma^2+(b-\Delta)(b-\psi)]^2[(1+\theta)(2-\gamma^2)-\theta(b-\Delta)(b-\psi)]^2}$$

### 4.2. Altruistic Reciprocity Scenario

Similar to fairness concern, according to the actual situation of whether the manufacturer is showing altruistic reciprocity and the E-platform's perception about the manufacturer's altruistic reciprocity, the information structure of the E-CLSC's fairness concern can be noted as below, in Case I–Case IV. However, as the E-platform is always completely rational, the response function of the E-platform and the recycler is the same as in the basic model—i.e., $p^*(w) = \frac{\alpha + (1-\gamma^2)w}{2-\gamma^2}$, $s^*(w) = \frac{\gamma(\alpha-w)}{2-\gamma^2}$ and $t^*(w) = \frac{(b-\psi)(\alpha-w)}{2-\gamma^2}$.

① Case I: The manufacturer is completely rational, and this information is symmetric. The manufacturer's decision problem is $\underset{w}{Max}\pi_m^{MS} = \pi_m = [w - c_m + t(\Delta - b)](\alpha - p + \gamma s)$, which is common knowledge in the E-CLSC. Case I is equivalent to a decentralized decision in the basic model, so the optimal solution in equilibrium is as follows:

$$w^{MS*} = \frac{\alpha[2-\gamma^2+2(b-\Delta)(b-\psi)]+c_m(2-\gamma^2)}{2[2-\gamma^2+(b-\Delta)(b-\psi)]}$$

$$p^{MS*} = \frac{\alpha[3-\gamma^2+2(b-\Delta)(b-\psi)]+c_m(1-\gamma^2)}{2[2-\gamma^2+(b-\Delta)(b-\psi)]}$$

$$s^{MS*} = \frac{\gamma(\alpha-c_m)}{2[2-\gamma^2+(b-\Delta)(b-\psi)]}$$

$$t^{MS*} = \frac{(\alpha-c_m)(b-\psi)}{2[2-\gamma^2+(b-\Delta)(b-\psi)]}$$

$(w^{MS*}, p^{MS*}, s^{MS*}, t^{MS*})$ is the optimal solution of equilibrium in Case I. We can calculate the profit as follows:

$$\pi_m^{MS*} = \frac{(\alpha-c_m)^2}{4[2-\gamma^2+(b-\Delta)(b-\psi)]}$$

$$\pi_e^{MS*} = \frac{(2-\gamma^2)(\alpha-c_m)^2}{8[2-\gamma^2+(b-\Delta)(b-\psi)]^2}$$

$$\pi_r^{MS*} = \frac{(b-\psi)^2(\alpha-c_m)^2}{8[2-\gamma^2+(b-\Delta)(b-\psi)]^2}$$

$$\pi_{sc}^{MS*} = \frac{(\alpha-c_m)^2[3(2-\gamma^2)+2(b-\Delta)(b-\psi)+(b-\psi)^2]}{8[2-\gamma^2+(b-\Delta)(b-\psi)]^2}$$

② Case II: The manufacturer is completely rational, but this information is asymmetric. The E-platform infers the manufacturer's decision problem as $\underset{w}{Max}u_m^{MA'} = \pi_m + \varphi\pi_e$, but the manufacturer's real decision problem is $\underset{w}{Max}\pi_m^{MA} = \pi_m = [w - c_m + t(\Delta - b)](\alpha - p + \gamma s)$. Under asymmetrical information, the E-platform thinks that the manufacturer's utility is $u_m^{MA'}$, and $\frac{\partial^2 u_m^{MA'}}{\partial w^2} = -\frac{(2-\varphi)(2-\gamma^2)+2(b-\Delta)(b-\psi)}{(2-\gamma^2)^2} < 0$, so the E-platform thinks the optimal wholesale price is $w^{MA'*} = \frac{\alpha[(1-\varphi)(2-\gamma^2)+2(b-\Delta)(b-\psi)]+c_m(2-\gamma^2)}{(2-\varphi)(2-\gamma^2)+2(b-\Delta)(b-\psi)}$. Then, the optimal decision of the E-platform and the recycler is

$$p^{MA*} = \frac{\alpha[1+(1-\varphi)(2-\gamma^2)+2(b-\Delta)(b-\psi)]+c_m(1-\gamma^2)}{(2-\varphi)(2-\gamma^2)+2(b-\Delta)(b-\psi)}$$

$$s^{MA*} = \frac{\gamma(\alpha-c_m)}{(2-\varphi)(2-\gamma^2)+2(b-\Delta)(b-\psi)}$$

$$t^{MA*} = \frac{(\alpha-c_m)(b-\psi)}{(2-\varphi)(2-\gamma^2)+2(b-\Delta)(b-\psi)}$$

As the manufacturer's real decision is $\underset{w}{Max}\pi_m^{MA} = \pi_m = [w - c_m + t(\Delta - b)](\alpha - p + \gamma s)$, the optimal wholesale price is $w^{MA*} = \frac{\alpha[2-\gamma^2+2(b-\Delta)(b-\psi)]+c_m(2-\gamma^2)}{2[2-\gamma^2+(b-\Delta)(b-\psi)]}$.

$(w^{MA*}, p^{MA*}, s^{MA*}, t^{MA*})$ is the optimal solution of equilibrium in Case II. We can calculate the profit as follows:

$$\pi_m^{MA*} = \frac{(\alpha - c_m)^2 \{(2-\varphi)(2-\gamma^2)[2-\gamma^2+2(b-\Delta)(b-\psi)]+2(b-\Delta)^2(b-\psi)^2\}}{2[2-\gamma^2+(b-\Delta)(b-\psi)][(2-\varphi)(2-\gamma^2)+2(b-\Delta)(b-\psi)]^2}$$

$$\pi_e^{MA*} = \frac{(2-\gamma^2)(\alpha-c_m)^2[(1-\varphi)(2-\gamma^2)+(b-\Delta)(b-\psi)]}{2[2-\gamma^2+(b-\Delta)(b-\psi)][(2-\varphi)(2-\gamma^2)+2(b-\Delta)(b-\psi)]^2}$$

$$\pi_r^{MA*} = \frac{(\alpha-c_m)^2(b-\psi)^2}{2[(2-\varphi)(2-\gamma^2)+2(b-\Delta)(b-\psi)]^2}$$

$$\pi_{sc}^{MA*} = \frac{(\alpha-c_m)^2[(3-2\varphi)(2-\gamma^2)+(b-\psi)^2+2(b-\Delta)(b-\psi)]^2}{2[(2-\varphi)(2-\gamma^2)+2(b-\Delta)(b-\psi)]^2}$$

③ Case III: The manufacturer shows altruistic reciprocity, and this information is symmetric; the manufacturer's decision is common knowledge in the E-CLSC, as below:

$$\underset{w}{Max} u_m^{LS} = \pi_m + \varphi\pi_e = [w - c_m + t(\Delta - b)](\alpha - p + \gamma s) + \varphi[(p-w)(\alpha - p + \gamma s) - \frac{1}{2}s^2]$$

It is easy to compute the optimal solution as follows by the backward induction method.

$$w^{LS*} = \frac{\alpha[(1-\varphi)(2-\gamma^2)+2(b-\Delta)(b-\psi)]+c_m(2-\gamma^2)}{(2-\varphi)(2-\gamma^2)+2(b-\Delta)(b-\psi)}$$

$$p^{LS*} = \frac{\alpha[1+(1-\varphi)(2-\gamma^2)+2(b-\Delta)(b-\psi)]+c_m(1-\gamma^2)}{(2-\varphi)(2-\gamma^2)+2(b-\Delta)(b-\psi)}$$

$$s^{LS*} = \frac{\gamma(\alpha-c_m)}{(2-\varphi)(2-\gamma^2)+2(b-\Delta)(b-\psi)}$$

$$t^{LS*} = \frac{(\alpha-c_m)(b-\psi)}{(2-\varphi)(2-\gamma^2)+2(b-\Delta)(b-\psi)}$$

$(w^{LS*}, p^{LS*}, s^{LS*}, t^{LS*})$ is the optimal solution of equilibrium in Case III. We can calculate the profit as follows:

$$\pi_m^{LS*} = \frac{(\alpha-c_m)^2[(1-\varphi)(2-\gamma^2)+(b-\Delta)(b-\psi)]}{[(2-\varphi)(2-\gamma^2)+2(b-\Delta)(b-\psi)]^2}$$

$$\pi_e^{LS*} = \frac{(2-\gamma^2)(\alpha-c_m)^2}{2[(2-\varphi)(2-\gamma^2)+2(b-\Delta)(b-\psi)]^2}$$

$$\pi_r^{LS*} = \frac{(\alpha-c_m)^2(b-\psi)^2}{2[(2-\varphi)(2-\gamma^2)+2(b-\Delta)(b-\psi)]^2}$$

$$\pi_{sc}^{LS*} = \frac{(\alpha-c_m)^2[(3-2\varphi)(2-\gamma^2)+(b-\psi)^2+2(b-\Delta)(b-\psi)]^2}{2[(2-\varphi)(2-\gamma^2)+2(b-\Delta)(b-\psi)]^2}$$

④ Case IV: The manufacturer shows altruistic reciprocity, but this information is asymmetric. The E-platform infers the manufacturer's decision problem as $\underset{w}{Max}\pi_m^{LA'} = \pi_m$, but the manufacturer's real decision problem is $\underset{w}{Max}u_m^{LA} = \pi_m + \varphi\pi_e$. The E-platform infers the manufacturer's optimal wholesale price as $w^{LA'*} = \frac{\alpha[2-\gamma^2+2(b-\Delta)(b-\psi)]+c_m(2-\gamma^2)}{2[2-\gamma^2+(b-\Delta)(b-\psi)]}$. Under asymmetrical information, the optimal decision of the E-platform and the recycler is

$$p^{LA*} = \frac{\alpha[3-\gamma^2+2(b-\Delta)(b-\psi)]+c_m(1-\gamma^2)}{2[2-\gamma^2+(b-\Delta)(b-\psi)]}$$

$$s^{LA*} = \frac{\gamma(\alpha-c_m)}{2[2-\gamma^2+(b-\Delta)(b-\psi)]}$$

$$t^{LA*} = \frac{(\alpha-c_m)(b-\psi)}{2[2-\gamma^2+(b-\Delta)(b-\psi)]}$$

The manufacturer's real optimal wholesale price is $w^{LA*} = \frac{\alpha[(1-\varphi)(2-\gamma^2)+2(b-\Delta)(b-\psi)]+c_m(2-\gamma^2)}{(2-\varphi)(2-\gamma^2)+2(b-\Delta)(b-\psi)}$ based on the response function of the E-platform and the recycler—i.e., $p^*(w)$, $s^*(w)$ and $t^*(w)$.

$(w^{LA*}, p^{LA*}, s^{LA*}, t^{LA*})$ is the optimal solution of equilibrium in Case IV. We can calculate the profit as follows:

$$\pi_m^{LA*} = \frac{(\alpha-c_m)^2\{(2-\gamma^2)[(1-\varphi)(2-\gamma^2)+(4-\varphi)(b-\Delta)(b-\psi)]+2(b-\Delta)^2(b-\psi)^2\}}{4[(2-\varphi)(2-\gamma^2)+2(b-\Delta)(b-\psi)][2-\gamma^2+(b-\Delta)(b-\psi)]^2}$$

$$\pi_e^{LA*} = \frac{(2-\gamma^2)(\alpha-c_m)^2[(2+\varphi)(2-\gamma^2)+2(b-\Delta)(b-\psi)]}{8[(2-\varphi)(2-\gamma^2)+2(b-\Delta)(b-\psi)][2-\gamma^2+(b-\Delta)(b-\psi)]^2}$$

$$\pi_r^{LA*} = \frac{(b-\psi)^2(\alpha-c_m)^2}{8[2-\gamma^2+(b-\Delta)(b-\psi)]^2}$$

$$\pi_{sc}^{LA*} \frac{(\alpha-c_m)^2[3(2-\gamma^2)+2(b-\Delta)(b-\psi)+(b-\psi)^2]}{8[2-\gamma^2+(b-\Delta)(b-\psi)]^2}$$

*4.3. Comparison and Analysis of Equilibrium Strategy*

**Property 1.** *The effect of the E-platform's fairness concern on the optimal solution of the E-CLSC is shown in Table 1.*

**Table 1.** The effect of the E-platform's fairness concern on the optimal solution.

| Cases | $w^*$ | $p^*$ | $s^*$ | $t^*$ | $\pi_m^*$ | $\pi_e^*$ | $\pi_r^*$ | $\pi_{sc}^*$ |
|-------|-------|-------|-------|-------|-----------|-----------|-----------|--------------|
| I | - | - | - | - | - | - | - | - |
| II | ↓ | ↓ | ↑ | ↑ | ↓ | ↑ | ↑ | ↑ |
| III | ↓ | - | - | - | ↓ | ↑ | - | - |
| IV | - | ↑ | ↓ | ↓ | ↓ | ↓ | ↓ | ↓ |

Note: "↑" denotes positive correlation, "↓"denotes negative correlation, and "-" means no correlation.

According to Property 1, the wholesale price decreases with the E-platform's fairness concern, and once the information is symmetrical, the E-platform's fairness concern has no effect on the retail price, service level and waste recycling rate. Under asymmetrical information, the retail price increases with the E-platform's real fairness concern but decreases with the fairness concern perceived by the manufacturer, and both the service level and the waste recycling rate decrease with the E-platform's real fairness concern but increase with the fairness concern perceived by the manufacturer.

Furthermore, when the E-platform has fairness concerns and the information is symmetrical, the manufacturer can improve the E-platform's profit by reducing the wholesale price and alleviate the E-platform's negative unfair utility. When the fairness concern information is asymmetric, the E-platform will increase the retail price and reduce the service level so as to obtain more profits, which will directly shrink the market demand and recovery rate and ultimately reduce the profits of all members in the E-CLSC. However, when the E-platform is completely rational and the information is asymmetrical, the profits of all members in the E-CLSC will be improved.

**Property 2.** *The effect of the manufacturer's altruistic reciprocity on the optimal solution of the E-CLSC is shown in Table 2.*

**Table 2.** The effect of the manufacturer's altruistic reciprocity on the optimal solution.

| Cases | $w^*$ | $p^*$ | $s^*$ | $t^*$ | $\pi_m^*$ | $\pi_e^*$ | $\pi_r^*$ | $\pi_{sc}^*$ |
|---|---|---|---|---|---|---|---|---|
| I | - | - | - | - | - | - | - | - |
| II | - | ↓ | ↑ | ↑ | ↑ | ↓ | ↑ | ↑ |
| III | ↓ | ↓ | ↑ | ↑ | ↓ | ↑ | ↑ | ↑ |
| IV | ↓ | - | - | - | ↓ | ↑ | - | - |

Note: "↑" denotes positive correlation, "↓"denotes negative correlation, and "-" means no correlation.

According to Property 2, the wholesale price decreases with the manufacturer's altruistic reciprocity; no matter whether the information is symmetrical or not, the retail price decreases, but both the service level and the recycling rate increase with the manufacturer's altruistic reciprocity as perceived by the E-platform.

Furthermore, when the E-platform believes that the manufacturer shows altruistic reciprocity, the E-platform will stimulate the market demand by reducing the retail price and improving the service level and encourage consumers to buy more products while improving the recycling rate of waste products; thus, both the E-platform and the manufacturer will improve the profits of both the recycler and the E-CLSC. When the manufacturer performs altruistic reciprocity and the information is asymmetric, the manufacturer can only improve the E-platform's profit by reducing the wholesale price, without changing the profit of recycler and the E-CLSC.

**Proposition 1.** ① $w^{NS*} = w^{FA*} > w^{NA*} = w^{FS*}$, $p^{FA*} > p^{NS*} = p^{FS*} > p^{NA*} > p^{C*}$, $s^{C*} > s^{NA*} > s^{NS*} = s^{FS*} > s^{FA*}$, $t^{C*} > t^{NA*} > t^{NS*} = t^{FS*} > t^{FA*}$; ② $w^{MS*} = w^{MA*} > w^{LS*} = w^{LA*}$, $p^{MS*} = p^{LA*} > p^{MA*} = p^{LS*} > p^{C*}$, $s^{C*} > s^{MA*} = s^{LS*} > s^{MS*} = s^{LA*}$, $t^{C*} > t^{MA*} = t^{LS*} > t^{MS*} = t^{LA*}$.

Proposition 1 ① and ② show that the retail price is the lowest and both the service level and the recycling rate are the highest in the centralized decision-making mode. No matter whether the information is symmetrical or not, as long as the manufacturer considers the E-platform's fairness concern or the E-platform considers the manufacturer's altruistic reciprocity to make a decision, it is conducive to reducing the wholesale price and retail price and improving the service level and the recycling rate of waste products. At the same time, no matter what kind of social preference is in the E-CLSC, information asymmetry is not conducive to reducing the retail price, improving the service level and promoting waste recycling.

Combined with Property 1, Property 2 and Proposition 1, no matter whether the social preference information is symmetrical, as long as each member considers the partner's social preference, it is conducive to reducing the retail price and improving the service level and the recycling rate of waste products.

**Proposition 2.** ① $\pi_m^{NS*} > \pi_m^{NA*} > \pi_m^{FS*} > \pi_m^{FA*}$, $\pi_e^{NA*} > \pi_e^{FS*} > \pi_e^{NS*} > \pi_e^{FA*}$, $\pi_r^{NA*} > \pi_r^{NS*} = \pi_r^{FS*} > \pi_r^{FA*}$, $\pi_{sc}^{C*} > \pi_{sc}^{NA*} > \pi_{sc}^{NS*} = \pi_{sc}^{FS*} > \pi_{sc}^{FA*}$; ② $\pi_m^{MA*} > \pi_m^{MS*} > \pi_m^{LS*} > \pi_m^{LA*}$, $\pi_e^{LS*} > \pi_e^{LA*} > \pi_e^{MS*} > \pi_e^{MA*}$, $\pi_r^{MA*} = \pi_r^{LS*} > \pi_r^{MS*} = \pi_r^{LA*}$, $\pi_{sc}^{C*} > \pi_{sc}^{MA*} = \pi_{sc}^{LS*} > \pi_{sc}^{MS*} = \pi_{sc}^{LA*}$.

Proposition 2 ① and ② show that no matter whether an E-CLSC member has a social preference or the information is symmetrical, the E-CLSC profit is always the highest. Proposition 2 ① illustrates that when the E-platform has fairness concerns but is not considered by the manufacturer, the profit of each member in the E-CLSC would reach the lowest level, so the manufacturer should consider the E-platform's fairness concerns to make decisions to improve the profits of all members.

Proposition 2 ① shows that when information is symmetrical, the E-platform's fairness concern can only regulate the profit between itself and the manufacturer ($\pi_m^{NS*} > \pi_m^{FS*}$, $\pi_e^{NS*} < \pi_e^{FS*}$) with no effect on the recycler and the E-CLSC ($\pi_r^{NS*} = \pi_r^{FS*}$, $\pi_{sc}^{NS*} = \pi_{sc}^{FS*}$).

Proposition 2 ② illustrates that if the E-platform considers the manufacturer's altruistic reciprocity to make a decision, the profits of both the recycler and the E-CLSC can be improved. When the information is asymmetric, the manufacturer's altruistic reciprocity can only adjust the profit between itself and the E-platform ($\pi_m^{MA*} > \pi_m^{LS*}$, $\pi_e^{MA*} < \pi_e^{LS*}$), with no effect on the profits of the recycler and the E-CLSC ($\pi_r^{MA*} = \pi_r^{LS*}$, $\pi_{sc}^{MA*} = \pi_{sc}^{LS*}$).

According to Propositions 1 and 2, Conclusion 1 can be obtained as below.

**Conclusion 1.** *Whether the social preference information is symmetrical or not, only the wholesale price contract cannot coordinate the E-CLSC, but as long as each member considers the partner's social preference to make a decision, it is conducive to optimizing decisions in the E-CLSC.*

## 5. Coordination Mechanism Based on Revenue-Sharing and Cost-Sharing Contract

In a revenue-sharing contract, a retailer can obtain products at a lower wholesale price before the sale season, but the retailer should share a certain proportion of the sale revenue with the supplier at the end of the sale season. In this way, the revenue-sharing contract can share profits and risks between the retailer and the supplier, and it is widely applied in leasing, audio–visual products, the Internet and other industries [39]. Similarly, a cost-sharing contract can share the cost and operation risk between supply chain members. According to our model structure, we applied a revenue-sharing contract between the manufacturer and the E-platform and applied a cost-sharing contract between the manufacture and the recycler, whereby the revenue-sharing and cost-sharing contract (shortened as RC contract) was designed to achieve optimal decision-making and E-CLSC coordination. In the RC contract, the manufacturer provides a lower wholesale price to reduce the E-platform's cost occupation, and the E-platform shares partial profits with the manufacturer at the end of the sale period to ensure the manufacturer's reasonable profit. At the same time, the manufacturer would share part of the recovery cost with the recycler, but the recycler should promise to achieve the optimal recovery rate of the E-CLSC. $\lambda(\vartheta)$ and $\mu(\phi)$ respectively express the revenue sharing proportion and the cost sharing proportion under the E-platform's fairness concern (manufacturer's altruistic reciprocity), $\lambda, \mu, \vartheta, \phi \in [0,1]$.

### 5.1. Fairness Concern Scenario

To ensure that the RC contract achieves supply chain coordination, let the E-platform's service be equal to optimal service in the centralized mode—i.e., $s^{RC*} = s^{C*}$. Then, the profit of each member can be denoted as below under the E-platform's fairness concern:

$$\pi_m^{RC} = [w - c_m + t(\Delta - b)](\alpha - p + \gamma s^{C*}) + (1 - \lambda)p(\alpha - p + \gamma s^{C*}) - \frac{1}{2}(1 - \mu)(t)^2 \quad (9)$$

$$\pi_e^{RC} = (\lambda p - w)(\alpha - p + \gamma s^{C*}) - \frac{1}{2}(s^{C*})^2 \quad (10)$$

$$\pi_r^{RC} = (b - \psi)(\alpha - p + \gamma s^{C*})t - \frac{1}{2}\mu(t)^2 \quad (11)$$

$$u_e^{RC} = (1 + \theta)[(\lambda p - w)(\alpha - p + \gamma s^{C*}) - \frac{1}{2}(s^{C*})^2] - \theta\{[w - c_m + t(\Delta - b)](\alpha - p + \gamma s^{C*})$$
$$+ (1 - \lambda)p(\alpha - p + \gamma s^{C*}) - \frac{1}{2}(1 - \mu)t^2\} \quad (12)$$

By calculating inequality constraints in each case, Conclusion 2 and Property 3 can be obtained.

**Conclusion 2.** *If $\mu^* = \frac{b-\psi}{\Delta-\psi}$, the RC contract can always achieve optimal decision-making and E-CLSC coordination under the E-platform's fairness concern, and the constraints should be met as follows:*

① $\lambda \in [\underset{\sim}{\lambda_I}, \widetilde{\lambda}_I]$, $w^{NS-RC*} = \frac{\lambda[c_m(2-\gamma^2) - \alpha(\Delta-\psi)^2]}{2 - \gamma^2 - (\Delta-\psi)^2}$

② $\quad \lambda \in [\underset{\sim}{\lambda}_{\text{II}}, \widetilde{\lambda}_{\text{II}}], w^{NA-RC*} = \dfrac{\lambda(1+2\theta)[c_m(2-\gamma^2)-\alpha(\Delta-\psi)^2]+\theta(\alpha-c_m)(\Delta-\psi)(b-\psi)}{(1+2\theta)[2-\gamma^2-(\Delta-\psi)^2]}$

③ $\quad \lambda \in [\underset{\sim}{\lambda}_{\text{III}}, \widetilde{\lambda}_{\text{III}}], w^{FS-RC*} = w^{NA-RC*}$

④ $\quad \lambda \in [\underset{\sim}{\lambda}_{\text{IV}}, \widetilde{\lambda}_{\text{IV}}], w^{FA-RC*} = w^{NS-RC*}.$

Here, $\underset{\sim}{\lambda}_{\text{I}} = \frac{1}{2}\gamma^2 + \dfrac{(2-\gamma^2)[2-\gamma^2-(\Delta-\psi)^2]^2}{8A^2}$,

$$\widetilde{\lambda}_{\text{I}} = 1 - \frac{1}{2}[(\Delta-\psi)^2+(\Delta-\psi)(b-\psi)] - \dfrac{[2-\gamma^2-(\Delta-\psi)^2]^2}{4A}$$

$$\underset{\sim}{\lambda}_{\text{II}} = \max\left\{\dfrac{\theta}{1+2\theta}, \frac{1}{2}\gamma^2 - \dfrac{\theta(\psi-\Delta)(b-\psi)}{1+2\theta} + \dfrac{B^2[2-\gamma^2-(\Delta-\psi)^2]^2}{8A^2(1+2\theta)^2(2-\gamma^2)}\right\},$$

$$\widetilde{\lambda}_{\text{II}} = 1 - \frac{1}{2}(\Delta-\psi)^2 + \dfrac{(\psi-\Delta)(b-\psi)}{2(1+2\theta)} - \dfrac{B(B-2\theta A)[2-\gamma^2-(\Delta-\psi)^2]^2}{4A(1+2\theta)}$$

$$\underset{\sim}{\lambda}_{\text{III}} = \max\left\{\dfrac{\theta}{1+2\theta}, \frac{1}{2}\gamma^2 - \dfrac{\theta(\psi-\Delta)(b-\psi)}{1+2\theta} + \dfrac{(B+\theta A)[2-\gamma^2-(\Delta-\psi)^2]^2}{8A^2(1+2\theta)}\right\},$$

$$\widetilde{\lambda}_{\text{III}} = 1 - \frac{1}{2}(\Delta-\psi)^2 + \dfrac{(\psi-\Delta)(b-\psi)}{2(1+2\theta)} - \dfrac{(1+\theta)[2-\gamma^2-(\Delta-\psi)^2]^2}{4A(1+2\theta)},$$

$$\underset{\sim}{\lambda}_{\text{IV}} = \frac{1}{2}\gamma^2 + \dfrac{(1+2\theta)(B-3\theta A)(2-\gamma^2)^2[2-\gamma^2-(\Delta-\psi)^2]^2}{8A^2(B-2\theta A)^2} \text{ and}$$

$$\widetilde{\lambda}_{\text{IV}} = 1 - \frac{1}{2}(\Delta-\psi)^2 + \frac{1}{2}(\psi-\Delta)(b-\psi) - \dfrac{(1+\theta)(2-\gamma^2)(B-3\theta A)[2-\gamma^2-(\Delta-\psi)^2]^2}{4A(B-2\theta A)^2}$$

Note $A = 2 - \gamma^2 + (b-\Delta)(b-\psi)$ and $B = (1+3\theta)(2-\gamma^2) + \theta(b-\Delta)(b-\psi)$.
Here, take the proof of Conclusion 2 ① as an example.

In Case I, we still adopt the backward induction method to solve the game equilibrium solution. In the second stage, $\frac{\partial^2 \pi_e^{NS-RC}}{\partial p^2} = -2\lambda < 0$, and the E-platform's optimal response function is

$$p^{NS-RC*}(w) = \dfrac{\lambda\alpha[2-(\Delta-\psi)^2]-\lambda c_m\gamma^2+w[2-\gamma^2-(\Delta-\psi)^2]}{2\lambda[2-\gamma^2-(\Delta-\psi)^2]}$$

For $\frac{\partial^2 \pi_r^{NS-RC}}{\partial t^2} = -\mu < 0$, the recycler's optimal response function is

$$t^{NS-RC*}(w) = \dfrac{(b-\psi)\lambda[2\alpha-\alpha(\Delta-\psi)^2-c_m\gamma^2]-(b-\psi)w[(2-\gamma^2)-(\Delta-\psi)^2]}{2\lambda\mu[2-\gamma^2-(\Delta-\psi)^2]}$$

When the RC contract realizes E-CLSC coordination, the E-CLSC profits under the decentralized decision mode will reach the maximal profits in the centralized decision mode. Given a constant unit cost, the optimal retail price and recycling rate should also equal those in the centralized decision mode—i.e., $p^{NS-RC*}(w) = p^{C*}$ and $t^{NS-RC*}(w) = t^{C*}$. We can obtain $\mu^* = \frac{b-\psi}{\Delta-\psi}$ and $w^{NS-RC*} = \dfrac{\lambda[c_m(2-\gamma^2)-\alpha(\Delta-\psi)^2]}{2-\gamma^2-(\Delta-\psi)^2}$ by solving two equations simultaneously.

Taking $\mu^*$ and $w^{NS-RC*}$ into Formulas (9)–(11), we can calculate the profits of the manufacturer, the E-platform, the recycler and the supply chain as below:

$$\pi_m^{NS-RC*} = \frac{(\alpha-c_m)^2[2(1-\lambda)-(\Delta-\psi)^2+(\psi-\Delta)(b-\psi)]}{2[2-\gamma^2-(\Delta-\psi)^2]^2}$$

$$\pi_e^{NS-RC*} = \frac{(2\lambda-\gamma^2)(\alpha-c_m)^2}{2[2-\gamma^2-(\Delta-\psi)^2]^2}$$

$$\pi_r^{NS-RC*} = \frac{(\Delta-\psi)(b-\psi)(\alpha-c_m)^2}{2[2-\gamma^2-(\Delta-\psi)^2]^2}$$

$$\pi_{sc}^{NS-RC*} = \frac{(\alpha-c_m)^2}{2[2-\gamma^2-(\Delta-\psi)^2]}$$

Furthermore, we should consider the individual rationality constraints of the E-CLSC members—i.e., $\pi_m^{NS-RC*} \geq \pi_m^{NS*}$, $\pi_e^{NS-RC*} \geq \pi_e^{NS*}$ and $\pi_r^{NS-RC*} \geq \pi_r^{NS*}$. For $\frac{\pi_r^{NS-RC*}}{\pi_r^{NS*}} = \frac{4(\Delta-\psi)[2-\gamma^2+(b-\Delta)(b-\psi)]^2}{(b-\psi)[2-\gamma^2-(\Delta-\psi)^2]^2} > 1$, $\pi_r^{NS-RC*} \geq \pi_r^{NS*}$ always holds, and we only consider constraints $\pi_m^{NS-RC*} \geq \pi_m^{NS*}$ and $\pi_e^{NS-RC*} \geq \pi_e^{NS*}$ as below:

$$\begin{cases} \frac{(\alpha-c_m)^2[2(1-\lambda)-(\Delta-\psi)^2+(\psi-\Delta)(b-\psi)]}{2[2-\gamma^2-(\Delta-\psi)^2]^2} \geq \frac{(\alpha-c_m)^2}{4[2-\gamma^2+(b-\Delta)(b-\psi)]} \\ \frac{(2\lambda-\gamma^2)(\alpha-c_m)^2}{2[2-\gamma^2-(\Delta-\psi)^2]^2} \geq \frac{(2-\gamma^2)(\alpha-c_m)^2}{8[2-\gamma^2+(b-\Delta)(b-\psi)]^2} \end{cases}$$

Thus, we can obtain

$$\frac{1}{2}\gamma^2 + \frac{(2-\gamma^2)[2-\gamma^2-(\Delta-\psi)^2]^2}{8A^2} \leq \lambda \leq 1 - \frac{1}{2}[(\Delta-\psi)^2 + (\Delta-\psi)(b-\psi)] - \frac{[2-\gamma^2-(\Delta-\psi)^2]^2}{4A}$$

Similarly, we can prove Conclusion 2 ②–④.

**Property 3.** *(1)* $\frac{\partial\lambda_{\mathrm{II}}}{\partial\theta} > 0$, $\frac{\partial\widetilde{\lambda}_{\mathrm{II}}}{\partial\theta} > 0$; *(2)* $\frac{\partial\lambda_{\mathrm{III}}}{\partial\theta} > 0$, $\frac{\partial\widetilde{\lambda}_{\mathrm{III}}}{\partial\theta} > 0$; *(3)* $\frac{\partial\lambda_{\mathrm{IV}}}{\partial\theta} < 0$, $\frac{\partial\widetilde{\lambda}_{\mathrm{IV}}}{\partial\theta} > 0$.

According to Property 3, both the upper and lower limits of the revenue sharing ratio increase with the E-platform's fairness concern in Cases II and III, which indicates that the stronger the E-platform's fairness concern is, the greater the revenue sharing ratio obtained is, which can effectively overcome the negative unfairness utility and achieve E-CLSC coordination. Additionally, the upper limit of the revenue sharing proportion increases with the E-platform's fairness concern, and the lower limit of the revenue sharing proportion decreases with the E-platform's fairness concern in Case IV. This shows that when the E-platform has fairness concerns and the information is asymmetric, the range of the revenue sharing ratio becomes larger, which makes it easier to achieve E-CLSC coordination and also reflects that the coordination and negotiation between the manufacturer and the E-platform can become less difficult.

### 5.2. Altruistic Reciprocity Scenario

Similar to the fairness concern scenario, let the E-platform's service equal the optimal service in the centralized mode, i.e., $s^{RC*} = s^{C*}$, to ensure the RC contract achieves supply chain coordination; thus, we can obtain the profit and utility of each member as below when the manufacturer performs altruistic reciprocity.

$$\pi_m^{RC} = [w - c_m + t(\Delta - b)](\alpha - p + \gamma s^{C*}) + (1 - \vartheta)p(\alpha - p + \gamma s^{C*}) - \frac{1}{2}(1 - \phi)t^2 \quad (13)$$

$$\pi_e^{RC} = (\vartheta p - w)(\alpha - p + \gamma s^{C*}) - \frac{1}{2}(s^{C*})^2 \quad (14)$$

$$\pi_r^{RC} = (b - \psi)(\alpha - p + \gamma s^{C*})t - \frac{1}{2}\phi t^2 \tag{15}$$

$$u_m^{RC} = [w - c_m + t(\Delta - b)](\alpha - p + \gamma s^{C*}) + (1 - \vartheta)p(\alpha - p + \gamma s^{C*}) - \frac{1}{2}(1 - \phi)t^2$$
$$+ \varphi[(\vartheta p - w)(\alpha - p + \gamma s^{C*}) - \frac{1}{2}(s^{C*})^2] \tag{16}$$

By calculating the inequality constraints in each case, Conclusion 3 and Property 4 can be obtained.

**Conclusion 3.** *If $\phi^* = \frac{b - \psi}{\Delta - \psi}$, the RC contract can always achieve optimal decision-making and E-CLSC coordination under the manufacturer's altruistic reciprocity, and the constraints should be met as follows:*

① *In Case I, $\vartheta \in [\underset{\sim}{\vartheta}_I, \widetilde{\vartheta}_I]$;* ② *In Case II, $\vartheta \in [\underset{\sim}{\vartheta}_{II}, \widetilde{\vartheta}_{II}]$;*
③ *In Case III, $\vartheta \in [\underset{\sim}{\vartheta}_{III}, \widetilde{\vartheta}_{III}]$;* ④ *In Case IV, $\vartheta \in [\underset{\sim}{\vartheta}_{IV}, \widetilde{\vartheta}_{IV}]$.*

*Here, $\underset{\sim}{\vartheta}_I = \frac{1}{2}\gamma^2 + \frac{(2-\gamma^2)[2-\gamma^2-(\Delta-\psi)^2]^2}{8A^2}$, $\widetilde{\vartheta}_I = 1 - \frac{1}{2}[(\Delta - \psi)^2 + (\Delta - \psi)(b - \psi)] - \frac{[2-\gamma^2-(\Delta-\psi)^2]^2}{4A}$*

$$\underset{\sim}{\vartheta}_{II} = \frac{1}{2}\gamma^2 + \frac{(K-A)(2-\gamma^2)[2-\gamma^2-(\Delta-\psi)^2]^2}{2AK^2},$$

$$\widetilde{\vartheta}_{II} = 1 - \frac{1}{2}[(\Delta - \psi)^2 + (\Delta - \psi)(b - \psi)] - \frac{[2-\gamma^2-(\Delta-\psi)^2]^2\left[(2-\varphi)(2-\gamma^2)[A+(b-\Delta)(b-\psi)]+2(b-\Delta)^2(b-\psi)^2\right]}{2AK^2}$$

$$\widetilde{\vartheta}_{III} = 1 - \frac{1}{2}[(\Delta - \psi)^2 + (\Delta - \psi)(b - \psi)] - \frac{(K-A)[2-\gamma^2-(\Delta-\psi)^2]^2}{K^2},$$

$$\underset{\sim}{\vartheta}_{IV} = \frac{1}{2}\gamma^2 + \frac{(2-\gamma^2)[2-\gamma^2-(\Delta-\psi)^2]^2[K+\varphi(2-\gamma^2)]}{8KA^2} \text{ and}$$

$$\widetilde{\vartheta}_{IV} = 1 - \frac{1}{2}[(\Delta - \psi)^2 + (\Delta - \psi)(b - \psi)] - \frac{[2-\gamma^2-(\Delta-\psi)^2]^2\left[(2-\gamma^2)[(4-\varphi)A-3(2-\gamma^2)]+2(b-\Delta)^2(b-\psi)^2\right]}{4KA^2}$$

*Note $K = (2 - \varphi)(2 - \gamma^2) + 2(b - \Delta)(b - \psi)$.*

**Property 4.** $\frac{\partial \underset{\sim}{\vartheta}_{II}}{\partial \varphi} < 0, \frac{\partial \widetilde{\vartheta}_{II}}{\partial \varphi} < 0; \frac{\partial \underset{\sim}{\vartheta}_{III}}{\partial \varphi} > 0, \frac{\partial \widetilde{\vartheta}_{III}}{\partial \varphi} > 0; \frac{\partial \underset{\sim}{\vartheta}_{IV}}{\partial \varphi} > 0, \frac{\partial \widetilde{\vartheta}_{IV}}{\partial \varphi} > 0.$

According to Property 4, both the upper and lower limits of the revenue sharing ratio decrease with the manufacturer's altruistic reciprocity as perceived by the E-platform. This illustrates that when the information is asymmetric, the stronger the manufacturer's altruistic reciprocity is perceived to be by the E-platform, the more profit the E-platform shares with the manufacturer under E-CLSC coordination.

Both the upper and lower limits of the revenue sharing ratio increase with the information intensity of the manufacturer's real altruistic reciprocity, which means that no matter whether the information is symmetrical, as long as the manufacturer's altruistic reciprocity becomes stronger, the E-platform can obtain a higher revenue sharing ratio, which can further increase the manufacturer's altruistic utility and promote E-CLSC coordination. For $\frac{\partial \underset{\sim}{\vartheta}_{III}}{\partial \varphi} > \frac{\partial \widetilde{\vartheta}_{III}}{\partial \varphi}, \left|\frac{\partial \widetilde{\vartheta}_{II}}{\partial \varphi}\right| > \left|\frac{\partial \underset{\sim}{\vartheta}_{II}}{\partial \varphi}\right|$, when the E-platform infers that the manufacturer is performing altruistic reciprocity, both the upper and lower limits of the revenue sharing ratio will gradually become smaller, and coordination between the manufacturer and the E-platform will become more difficult, which is not conducive to E-CLSC coordination, which also reflects that coordination and negotiation between the manufacturer and the E-platform become more difficult.

According to Conclusions 3 and 4, we can obtain Corollary 1.

**Corollary 1.** *No matter whether the information is symmetrical or not, social preference has no effect on the cost sharing ratio under E-CLSC coordination.*

According to Conclusions 3 and 4, when the RC contract coordinates the E-CLSC, the optimal cost sharing ratio is $\mu^* = \frac{b-\psi}{\Delta-\psi}$ under the E-platform's fairness concern and $\phi^* = \frac{b-\psi}{\Delta-\psi}$ under the manufacturer's altruistic reciprocity, and it is easy to find that $\mu^* = \phi^* = \frac{b-\psi}{\Delta-\psi}$ has no relation with the E-platform's fairness concern nor with the manufacturer's altruistic reciprocity. Whether the E-platform cares about the manufacturer's profit or the manufacturer cares about improving the E-platform's profit, as long as the manufacturer shares a constant cost ratio with the recycler, the RC contract could achieve E-CLSC coordination by adjusting the revenue sharing ratio with a different wholesale price. Specifically, $b - \psi$ denotes the recycler's unit recycling profit, and $\Delta - \psi$ denotes the manufacturer's unit remanufacturing profit. $\mu^* = \phi^* = \frac{b-\psi}{\Delta-\psi}$ reflects that the profit obtaining ratio is equal to the cost saving ratio. When the manufacturer decides the repurchase price of waste products for the recycler, the recycler will maintain a stable recycling rate of waste products, and thus, the recycler's profit will not change, and the cooperation between the recycler and the manufacturer will be stable.

## 6. Numerical Analysis

In order to analyze the effect of the E-platform's fairness concern and the manufacturer's altruistic reciprocity on the E-CLSC, we applied a numerical analysis by Maple to verify our conclusions, and the relative parameters can be assumed as $\Delta = 7$, $b = 6.8$, $\psi = 6.5$, $\gamma = 0.7$, $\alpha = 12$ and $c_m = 10$, which can meet all constraints.

### 6.1. Fairness Concern Scenario

Figures 2a–d and 3a–d can verify Property 1, Proposition 1 ① and Proposition 2 ①. According to Figure 3a–d, once the information is symmetric, the E-platform's fairness concern can only play the role of a "profit adjustment mechanism" between the manufacturer and the E-platform, with no effect on the recycler and the supply chain. Furthermore, in Figure 3d, no matter whether the E-platform has fairness concerns or the information is symmetric, just the wholesale price cannot coordinate the supply chain, which partially verifies Conclusion 1 under fairness concern.

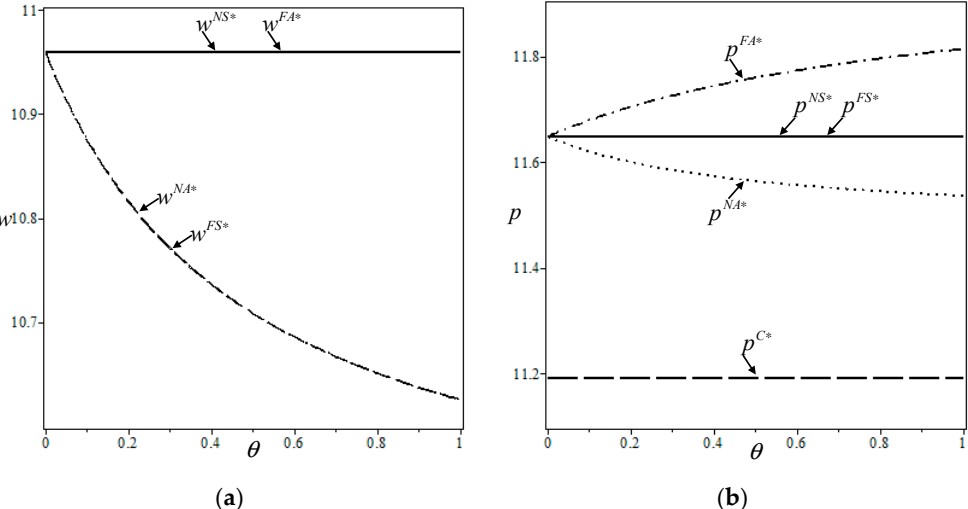

(a)    (b)

**Figure 2.** *Cont.*

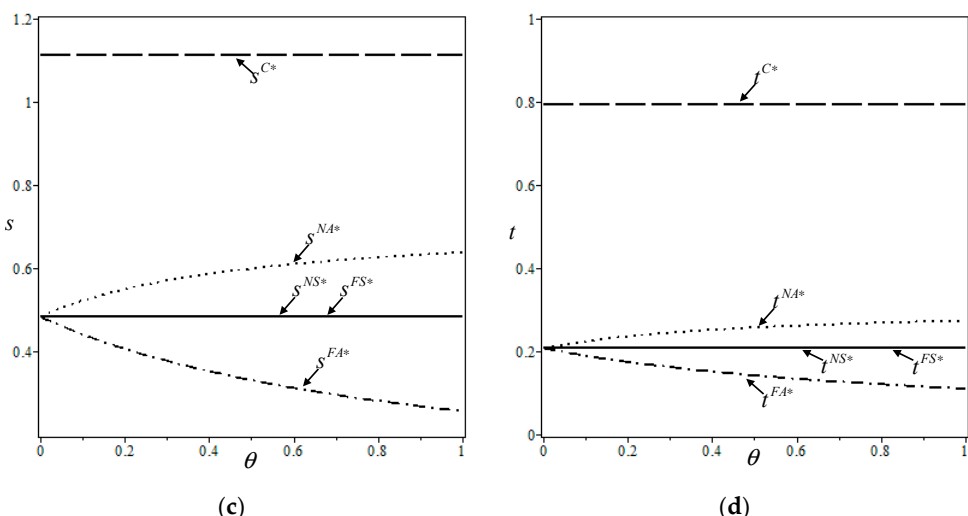

**Figure 2.** (**a**) Wholesale price. (**b**) Retail price. (**c**) E-platform's service. (**d**) Recycling rate. Under fairness concern Scenario.

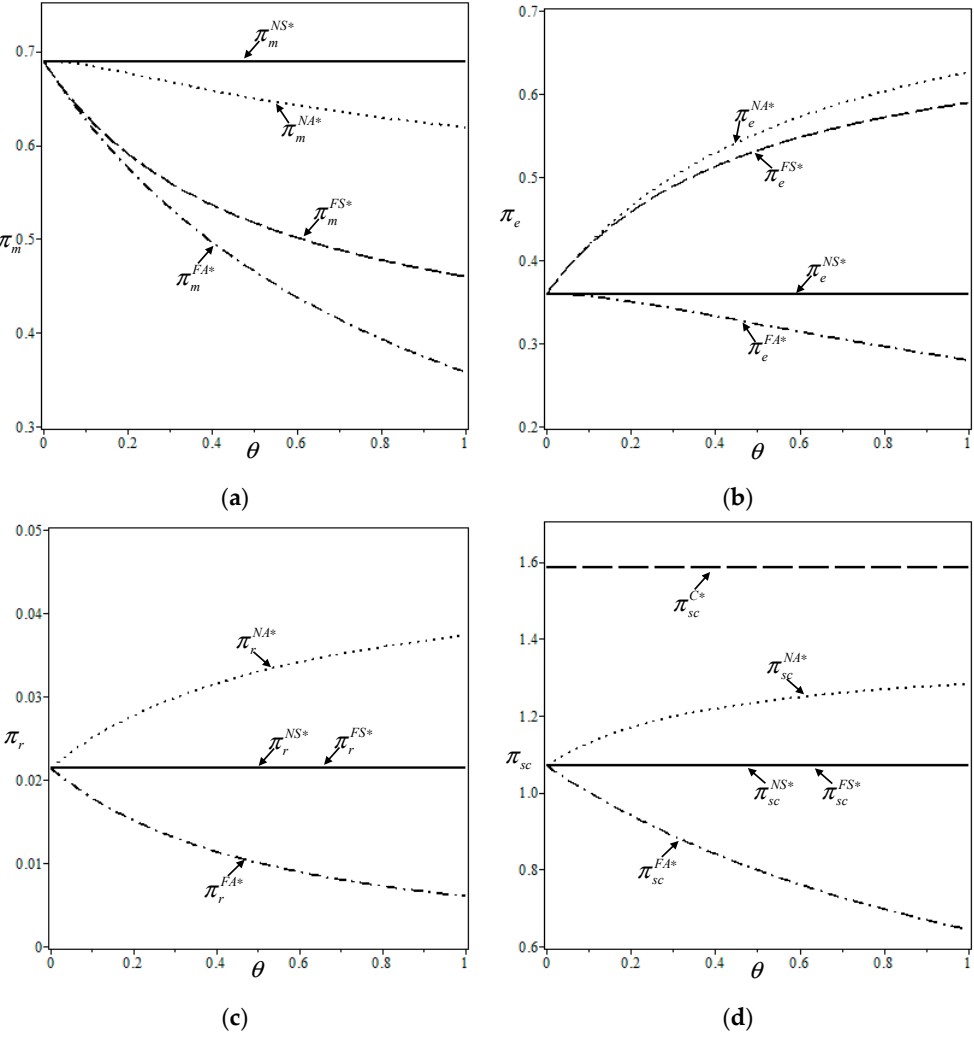

**Figure 3.** (**a**) Manufacturer's profit. (**b**) E-platform's profit. (**c**) Recycler's profit. (**d**) E-CLSC's profit. Under fairness concern scenario.

Figure 4a–d can verify Conclusion 2 and Property 3. Specifically, in Figure 4a, both the upper and lower limits of the revenue sharing ratio increase with the E-platform's fairness concern in Case II and Case III, whereas in Case IV, the upper limit of the revenue sharing proportion increases with the E-platform's fairness concern and the lower limit of the revenue sharing proportion decreases. This shows that when the E-platform shows fairness concerns and the information is asymmetric, the range of the revenue sharing proportion becomes larger, which makes it easier to achieve E-CLSC coordination and also reflects that the coordination and negotiation between the manufacturer and the E-platform can become less difficult. In particular, when the cost sharing ratio is 0.6 and the revenue sharing ratio is subjected to $\underset{\sim}{\lambda}_{II} < \lambda < \widetilde{\lambda}_{I}$, the dark area is the common coordination interval in four cases and $\theta = 0.748$ is the maximum value in the common coordination boundary. We drew the profit before and after the RC contract to obtain the comparison in $\theta \in [0, 0.6]$ as an example to illustrate Conclusion 2, as shown in Figure 4b–d.

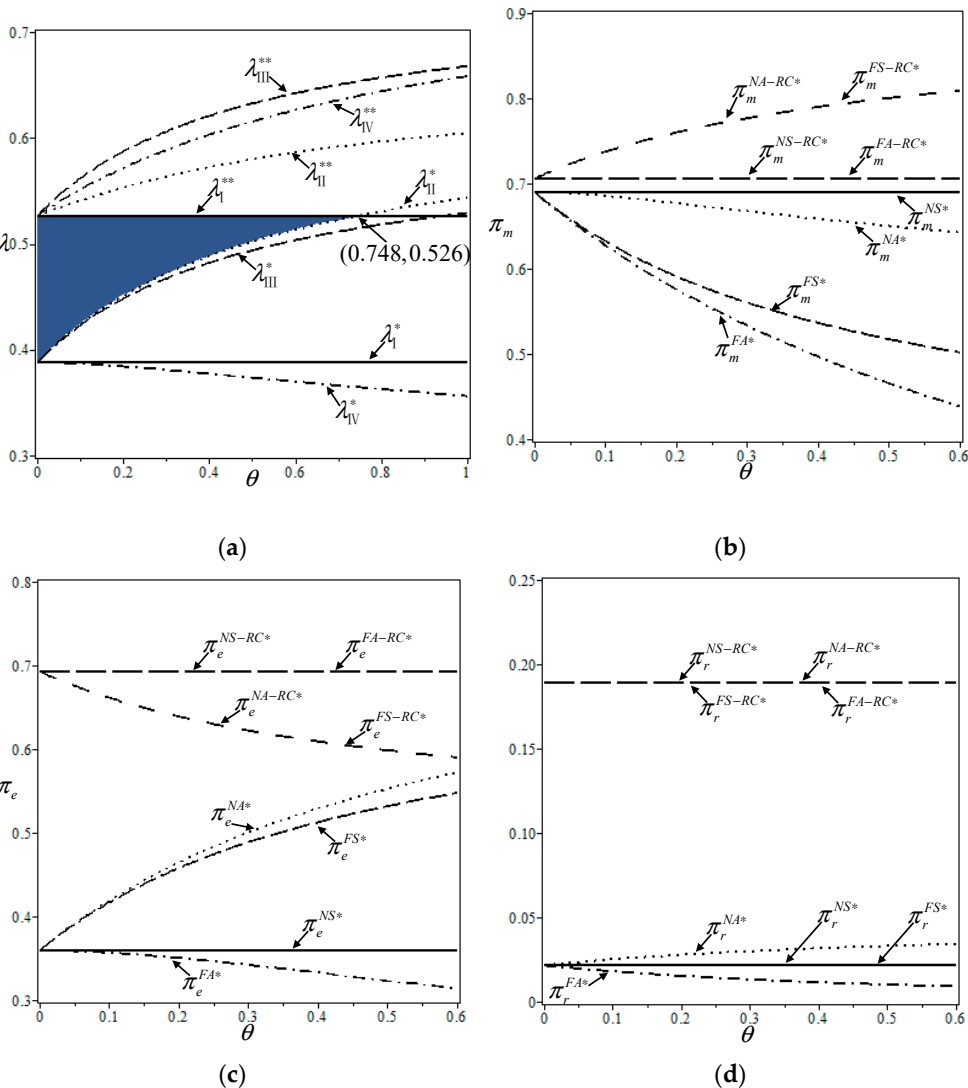

**Figure 4.** (**a**) Revenue sharing ratio. (**b**) Manufacturer's profit comparison. (**c**) E-platform's profit comparison. (**d**) Recycler's profit comparison. Under Fairness concern scenario.

### 6.2. Altruistic Reciprocity Scenario

Figures 5a–d and 6a–d can verify Property 2, Proposition 1 ② and Proposition 2 ②. According to Figure 6a–d, once the information is asymmetric, the manufacturer's altruistic reciprocity can only play the role of a "profit adjustment mechanism" between the manufacturer and the E-platform, with no effect on the recycler and the supply chain. Furthermore, in Figure 6d, no matter whether the manufacturer performs altruistic reciprocity or the information is symmetric, just the wholesale price cannot coordinate the supply chain, which partially verifies Conclusion 2 under altruistic reciprocity.

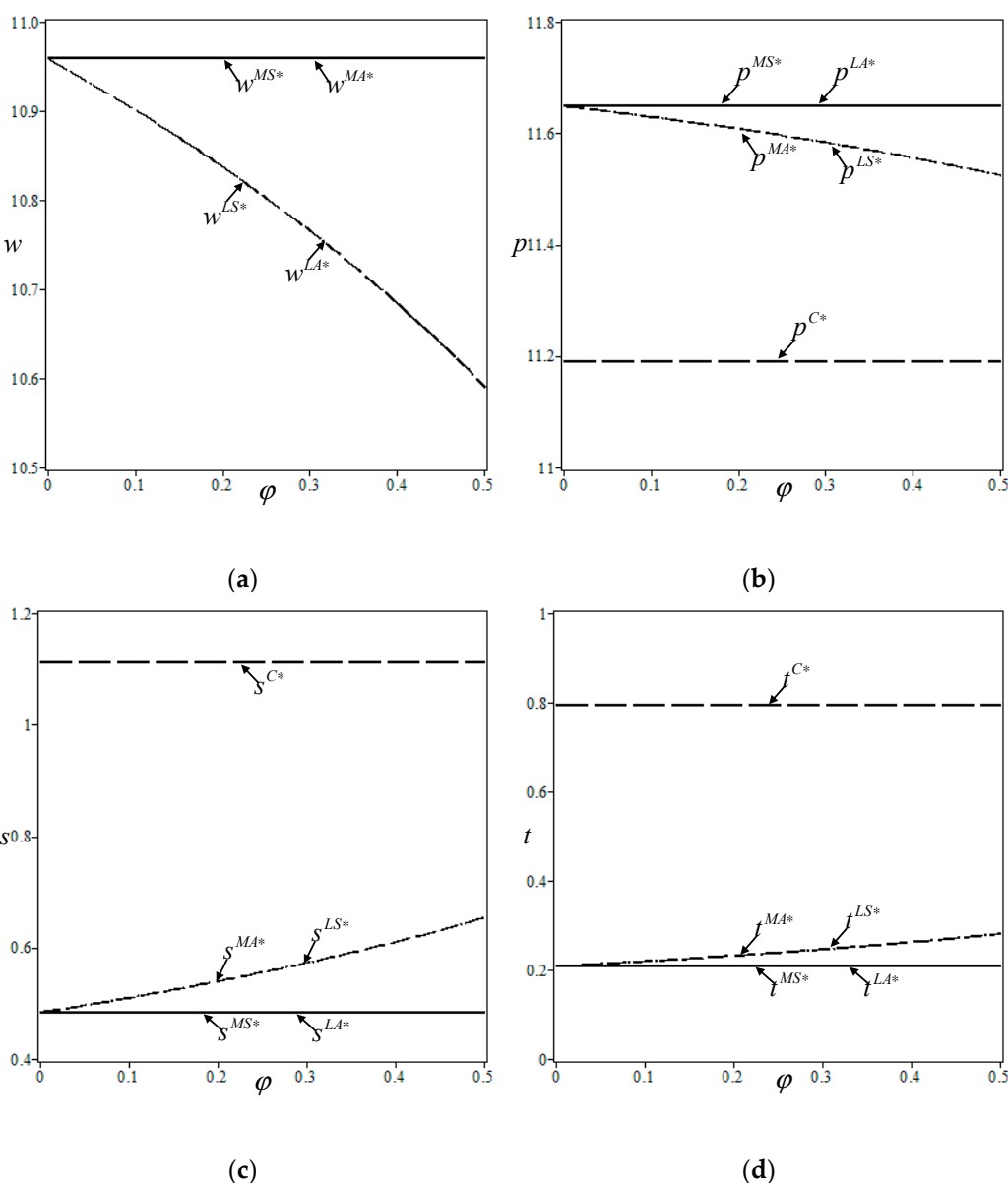

**Figure 5.** (**a**) Wholesale price. (**b**) Retail price. (**c**) E-platform's service. (**d**) Recycling rate. Under altruistic reciprocity scenario.

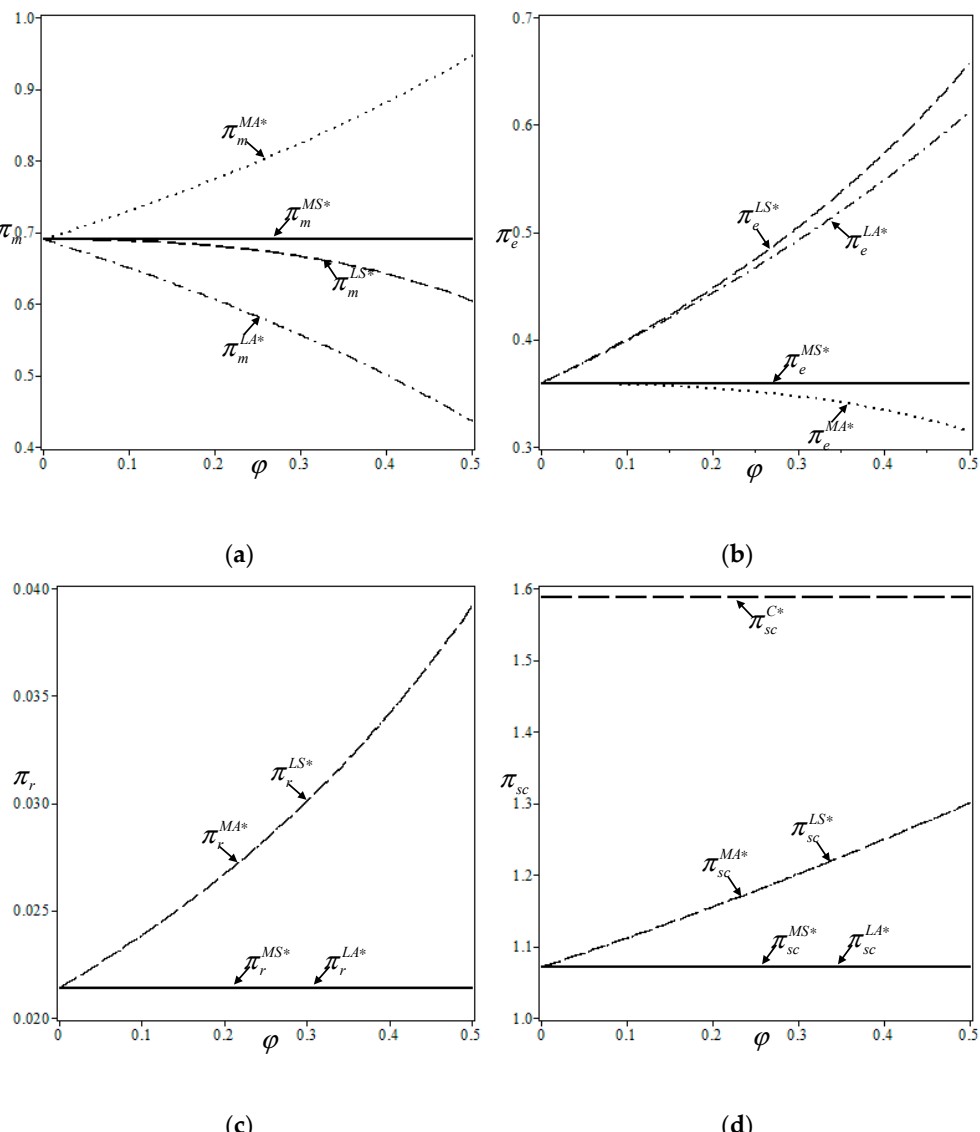

**Figure 6.** (**a**) Manufacturer's profit. (**b**) E-platform's profit. (**c**) Recycler's profit. (**d**) E-CLSC's profit. Under altruistic reciprocity scenario.

Figure 7a–d can verify Conclusion 3 and Property 4. Specifically, in Figure 7a, both the upper and lower limits of the revenue sharing ratio increase with the manufacturer's altruistic reciprocity in Case III and Case IV but decrease in Case II. At the same time, both the upper and lower limits of the revenue sharing ratio decrease, so the manufacturer's altruistic reciprocity would increase the coordination difficulty between itself and the E-platform. In particular, when the cost sharing ratio is 0.6 and the revenue sharing ratio is subjected to $\vartheta_{III} < \vartheta < \widetilde{\vartheta}_{II}$, the dark area is the common coordination interval in four cases and $\vartheta = 0.358$ is the maximum value in the common coordination boundary. We drew the profit before and after the RC contract to obtain the comparison in $\vartheta \in [0, 0.3]$ as an example to illustrate Conclusion 3, as shown in Figure 7b–d.

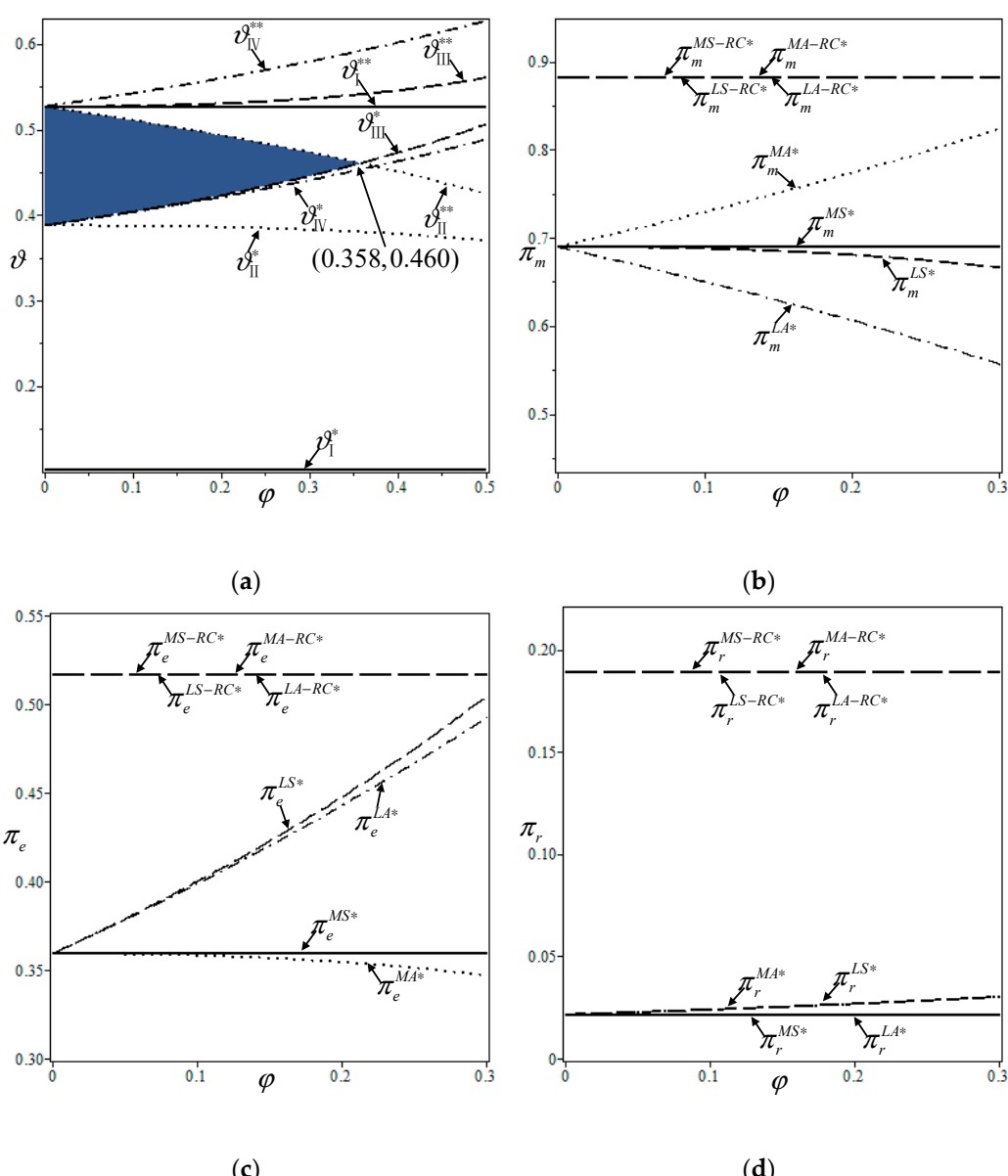

**Figure 7.** (**a**) Revenue sharing ratio. (**b**) Manufacturer's profit comparison. (**c**) E-platform's profit comparison. (**d**) Recycler's profit comparison. Under altruistic reciprocity scenario.

## 7. Conclusions

In this paper, we proposed an information structure to depict four symmetrical and asymmetrical cases of social preference—i.e., the E-platform's fairness concern and the manufacturer's altruistic reciprocity in the E-CLSC—and the backward induction method was adopted to solve the equilibrium in each case. By comparative analysis, the effect of the E-platform's fairness concern and the manufacturer's altruistic reciprocity on decision-making and coordination was analyzed, and the RC contract was proposed to coordinate the E-CLSC. We proved that as long as each member considers the partner's social preference to make a decision, it is conducive to optimizing the decision for all members and the E-CLSC. Secondly, when the E-platform has fairness concerns under symmetrical information or the manufacturer performs altruistic reciprocity under asymmetrical information, the E-CLSC social preference can only play the role of a profit distribution mechanism between the manufacturer and the E-platform, with no effect on the recycler and the supply chain. Thirdly, whether the social preference information is symmetrical or not, the wholesale price contract alone cannot coordinate the E-CLSC, but the RC contract can always achieve

optimal recycling decisions, coordinate the supply chain and Pareto-improve all parties' profits with a constant cost sharing ratio. Finally, the E-platform's fairness concern may widen the range of the revenue sharing ratio and make it easier to coordinate the E-CLSC, but the manufacturer's altruistic reciprocity may narrow the range of the revenue sharing ratio and make it harder to coordinate the E-CLSC.

From our research conclusions, we can suggest that E-CLSC members should consider each other's social preferences to make decisions so as to promote product sales, recycling rates and E-CLSC profits. For the manufacturer, the implementation of altruistic reciprocity is beneficial to E-CLSC development. Therefore, the manufacturer should properly implement altruistic measures to achieve mutual benefits and win–win results. For the E-platform, if it hides or exaggerates its fairness concern information, this will have a negative impact on the profit of all members in the E-CLSC, so the E-platform should actively transmit fairness concern information to promote information sharing among members. For the recycler, the revenue-sharing and cost-sharing contract can effectively coordinate the E-CLSC after the cost sharing proportion is determined, so the recycler should clarify the responsibility for cost sharing, promote cooperation with the manufacturer and improve the recycling rate of waste products.

There are some limitations, as below.

Firstly, in this paper, we only investigated the effect of the E-platform's fairness concern and the manufacturer's altruistic reciprocity on recycling decision optimization and contract coordination in an E-CLSC dominated by the manufacturer. In an E-CLSC that relies on an E-platform to recycle waste products and sell remanufactured products, it horizontally integrates all production processes, recycling processes and sale processes by network and achieves parallel operation of forward logistics, reverse logistics and sales processes, thus improving the efficiency of waste products' recycling and remanufacturing obviously. With the development of various intelligent network technologies (e.g., big data, cloud computing, blockchain, etc.), e-commerce will become more popular and E-platforms will thus play a more important role in every E-CLSC process, including manufacture, remanufacture, recycling and product sales. Therefore, E-platforms will have a key role in the operation of entire E-CLSCs, such as JD Mall, Tmall Global, Vipshop, Suning E-shop, etc., and will dominate E-CLSCs with a stronger pricing priority ability and strength than small manufacturers, so these large-scale E-platforms will occupy a dominant position in E-CLSCs, which are becoming E-platform-dominated. It is necessary to consider the E-platform's altruistic reciprocity and the manufacturer's fairness concern in an E-CLSC dominated by the E-platform, investigate the effect of social preferences on the decision-making and contract coordination and check whether the RC contract can coordinate the E-CLSC.

Secondly, our model included a manufacturer, an E-platform and a recycler, and the E-platform only cared about the upstream manufacturer's profits while ignoring the comparison with other competitive E-platforms' profits. Competitive E-platforms have direct competitiveness and are more likely to care about profit comparisons with other E-platforms in the same market position with similar services. Therefore, it is necessary to expand the social preference to the E-platform's competing condition in the E-CLSC, including multiple E-platforms. Although the complexity and difficulty of calculation will greatly increase, it is closer to the real E-CLSC competition environment. For example, we can set another model including a manufacturer and two competitive E-platforms, and it is necessary to consider a profit comparison between each E-platform and the manufacturer and a profit comparison between the various E-platforms so as to investigate the effect of social preference on the recycling decision and contract coordination and check whether flexible contracts (e.g., revenue-sharing and cost-sharing contract, transfer payment contract, two-part pricing contract, etc.) can coordinate the E-CLSC. It is also important to check how social preference influences the contract coordination range so as to propose more practical management strategies for improving recycling efficiency and E-CLSC operation.

Finally, we investigated the impact of the E-platform's fairness concern and the manufacturer's altruistic reciprocity on the recycling decision and contract coordination of the E-CLSC under symmetrical and asymmetrical information conditions. It is essential to identify the types and intensity of social preferences so as to avoid the negative impact of asymmetrical social preferences on the recycling decision. For example, we can apply a cost–benefit analysis to choose an E-platform to join the supply chain only when the revenue is greater than the relative cost.

**Author Contributions:** Conceptualization, Y.Q. and S.W.; methodology, N.G.; software, N.G.; validation, S.W.; formal analysis, Y.Q. and S.W.; investigation, Y.Q. and S.W.; resources, N.G.; writing—original draft preparation, S.W.; writing—review and editing, Y.Q. and S.W.; supervision, Y.Q.; project administration, Y.Q.; funding acquisition, Y.Q. All authors have read and agreed to the published version of the manuscript.

**Funding:** The research was supported by the Chongqing Social Science Major Planning Project (No. 2021ZDSC09), the Chongqing Humanities and Social Sciences Planning Project (No. 20SKGH296) and the Science Research Project of Chongqing Education Commission (KJQN202001123).

**Institutional Review Board Statement:** Not applicable.

**Informed Consent Statement:** Not applicable.

**Data Availability Statement:** Not applicable.

**Conflicts of Interest:** The authors declare no conflict of interest.

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
