# Peer review of "Coordination Mechanism of E-Closed-Loop Supply Chain under Social Preference"

_sustainability, doi:10.3390/su142013654_

Round 1

Reviewer 1 Report

This study investigates the implication of social preference on the recycling decision and coordination in E-closed-loop supply chain (E-CLSC). They prove that whether social preference information is symmetric or not, only wholesale price contract can’t coordinate E-CLSC, but the revenue-sharing and cost-sharing contract can always achieve optimal recycling decision, coordinate supply chain, and Pareto improve all parties’ profit with constant cost-sharing ratio. In addition, the E-platform’s fairness concern can widen the range of revenue-sharing ratio and make it easier to coordinate E-CLSC, but the manufacturer’s altruistic reciprocity may narrow the range of revenue-sharing ratio and make it harder to coordinate E-CLSC.

This paper has focused on interested area and has been written clearly. I believe that this piece of work can make an important contribution. However, there is some critical issue related to manuscript format. Here is my comments for the manuscript.

 1. Introduction & Literature review: After reading the two parts, it is not clear which is your research question and what you want to do. It is true you propose and point out several things but at the end I was a bit lost. I think it could be important to organize your ideas better and this will help your paper a lot! (Importance, gap, research question, objectives, contributions, structure). Specially, the revenue-sharing and cost-sharing contract is applied to coordinate supply chain, but there is no research referred to this contract. The literature about revenue-sharing and cost-sharing contract should be updated and latest references should be included.

 2.Why did the authors choose information structure to depict the four symmetry and asymmetry cases of social preference?

 3. Like other research, this one is also not perfect, so please include limitations of the study and the contributions of the paper should be summarized in the conclusion.

 4. There are some long sentences with multiple conjunctions in this paper, which makes them hard to understand by reader. I suggest the authors should check all of the long sentences throughout the manuscript. Breaking down long sentences into 2 simple sentences will be desired.

 5. The manuscript should be proofread in order to overcome the language problems and grammatical mistakes.

Author Response

Referees Response Letter

Thank for the comments to improve our paper, and we revised the paper according to the comments as following:

This study investigates the implication of social preference on the recycling decision and coordination in E-closed-loop supply chain (E-CLSC). They prove that whether social preference information is symmetric or not, only wholesale price contract can’t coordinate E-CLSC, but the revenue-sharing and cost-sharing contract can always achieve optimal recycling decision, coordinate supply chain, and Pareto improve all parties’ profit with constant cost-sharing ratio. In addition, the E-platform’s fairness concern can widen the range of revenue-sharing ratio and make it easier to coordinate E-CLSC, but the manufacturer’s altruistic reciprocity may narrow the range of revenue-sharing ratio and make it harder to coordinate E-CLSC.

This paper has focused on interested area and has been written clearly. I believe that this piece of work can make an important contribution. However, there is some critical issue related to manuscript format. Here is my comments for the manuscript.

  1. Introduction & Literature review: After reading the two parts, it is not clear which is your research question and what you want to do. It is true you propose and point out several things but at the end I was a bit lost. I think it could be important to organize your ideas better and this will help your paper a lot! (Importance, gap, research question, objectives, contributions, structure). Specially, the revenue-sharing and cost-sharing contract is applied to coordinate supply chain, but there is no research referred to this contract. The literature about revenue-sharing and cost-sharing contract should be updated and latest references should be included.

Response:

  1. we expanded our literature review, and we denote expanded literature review in green color in literature review, the detailed information is as following:

(1) In Page 3, line 137-140, Wang et al. (2022) study E-CLSC recycling decision under the financial constraint of the E-platform, and they prove that the financial constraint of the E-platform would have "win-win" effect on E-platform and recycler, and propose strategies and suggestions conducive to promoting the waste products recycle with case study [19].

(2) In Page 4, line 156-158, Qunar.com, basically because the price and service provided by Qunar.com were unreasonable, which led to unfair psychology of the partners, affected their direct profit, and damaged the balance of the overall supply chain operation [20].

(3) In Page4, line 198-202, Ding et al. (2022) study the impact of altruistic reciprocity on the recovery efficiency and system profit distribution of the closed-loop supply chain based on the scale diseconomies, and prove that whether recovery efficiency can be improved is depended on the supplier’s altruistic-reciprocity intensity and the manufacturer's attitude towards supplier’s altruistic reciprocity [34].

(4) In Page 5, line 208-211, Wang et al. (2022) design a contract of “quantity discount combined with fixed cost sharing” to achieve E-CLSC coordination by considering the effect of dominant manufacturer’s altruistic reciprocity on recycling decision under government incentive mechanism [37].

We add the detailed literatures in the References in green color as below:

[1] Wang, Y. Y., Su, M., Wang, X. D. Altruistic decision-making of closed-loop supply chain under government incentive mechanism, Chinese Journal of Management Science, In press, 2022.

[2] Wang, Y. Y., Yu, Z., Shen, L. Recycling decision of e-commerce closed-loop supply chain under capital constraint of e-commerce platform. Chinese Journal of Management Science, 2022,30(3): 154-164.

[3] Ding, J. F., Chen, W. D., Fu S S. The impact of reciprocal preferences on closed-loop supply chain with diseconomies of scale, Journal of Industrial Engineering/ Engineering Management, 2022, 36(1):194-204.

[4] Ding, J. F., Chen, W. D., Fu S S. The impact of reciprocal preferences on closed-loop supply chain with diseconomies of scale, Journal of Industrial Engineering/ Engineering Management, 2022, 36(1):194-204.

 2.Why did the authors choose information structure to depict the four symmetry and asymmetry cases of social preference?

The information structure can simply and clearly reflect four scenario of asymmetric social preference information

  1. Like other research, this one is also not perfect, so please include limitations of the study and the contributions of the paper should be summarized in the conclusion.

We add the contributions in green color in Page 5, in line 236-257, and limitations with green color in the Conclusion in Page 25, line 831-871, as below:

Our contribution lies in four aspects as below:

Firstly, we set the E-CLSC model including manufacture, E-platform and recycler, and investigate the recycling decision and contract coordination to meet the development of e-commerce and big data technology, optimize recycling decision and promote E-CLSC sustainable operation.

 Secondly, we investigate the impact of two typical social preference (i.e. fairness concern and altruistic reciprocity) on the recycling decision and contract coordination in E-CLSC more in line with the actual decision-making psychology, and thus can provide new analytical perspective for improving the recycling rate, promoting E-CLSC operation and recovering resources.

Thirdly, we analyze the impact of social preference as subjective private information on recycling decision and contract coordination in E-CLSC by depicting four cases of symmetric and asymmetric information under fairness concern and altruistic reciprocity with information structure, and prove that no matter whether social preference information is symmetric or not, wholesale price contract cannot coordinate the E-CLSC, but once the decision maker can consider the partner’s social preference to make decision, it is conducive to optimizing decision of all parties and supply chain.

Finally, we design revenue-sharing and cost-sharing contract to achieve the optimal recycling decision, supply chain coordination, Pareto improvement of each member's profit regardless of whether the information is symmetric or not. We prove that the E-platform’s fairness concern will enlarge the range of revenue sharing proportion and reduce the coordination difficulty of E-CLSC, while the manufacturer's altruistic reciprocity will shrink the range of revenue sharing proportion and increase the coordination difficulty of E-CLSC.

There are some limitations as below:

Firstly, in this paper, we only investigate the effect of E-platform’s fairness concern and manufacturer’s altruistic reciprocity on recycling decision optimization and contract coordination in the E-CLSC dominated by manufacturer. For E-CLSC relies on E-platform to recycle waste products and sell remanufactured products, horizontally integrates all production process, recycling process and sale process by network, and achieves the parallel operation of forward logistics, reverse logistics and sales processes so as to improve the efficiency of waste products recycling and remanufacturing obviously. With the development of various intelligent network technologies (e.g. big data, cloud computing, blockchain, etc.), e-commerce will become more popular and thus E-platform will play a more important role in every E-CLSC process including manufacture, remanufacture, recycle and product sale, so E-platform will become a key role in the operation of entire E-CLSC, e.g. JD Mall, Tmall Global, Vipshop, Suning E-shop, etc., and they will dominate E-CLSC with stronger pricing priority ability and strength than small manufacturers, so these large-scale E-platforms usually occupy a dominant position in E-CLSC and E-CLSC is becoming E-platform dominated. It is necessary to consider the E-platform’s altruistic reciprocity and manufacturer’s fairness concern in E-CLSC dominated by E-platform, and investigate the effect of social preference on the decision-making and contract coordination, and check whether the RC contract can coordinate E-CLSC.

Secondly, our model includes a manufacturer, an E-platform and a recycler, and E-platform only cares about the upstream manufacturer’s profit, while ignoring the profit comparison with other competitive E-platforms’ profit. The competitive E-platforms have direct competitiveness and are more likely to care about the profit with each other in the same market position with similar service. Therefore, it is necessary to expand the social preference to E-platforms competing condition in the E-CLSC including multiple E-platforms. Although the complexity and difficulty of calculation will greatly increase, it is closer to the real E-CLSC competition environment. For example, we can set another model including a manufacturer and two competitive E-platforms, and it is necessary to consider profit comparison between E-platform and manufacturer and profit comparison between various E-platforms so as to investigate the effect of social preference on the recycling decision and contract coordination, and check whether flexible contracts (e.g. revenue-sharing and cost-sharing contract, transfer payment contract, two-part pricing contract, etc.) can coordinate E-CLSC and how social preference influence the contract coordination range, so as to propose more practical management strategies for improving recycling efficiency and E-CLSC operation.

Finally, we investigate the impact of E-platform’s fairness concern and manufacture’s altruistic reciprocity on the recycling decision and contract coordination of E-CLSC under symmetry and asymmetry information condition. It is essential to identify the types and intensity of social preference so as to avoid the negative impact of asymmetric social preference on the recycling decision. For example, we can apply the cost-benefit analysis to choose the E-platform to join supply chain only when the revenue is greater than relative cost.

  1. There are some long sentences with multiple conjunctions in this paper, which makes them hard to understand by reader. I suggest the authors should check all of the long sentences throughout the manuscript. Breaking down long sentences into 2 simple sentences will be desired.

We have read the full manuscript, and try to divide the long sentences into short sentences, which are marked in red in the paper.

  1. The manuscript should be proofread in order to overcome the language problems and grammatical mistakes.

We revised the language problems and grammatical mistakes in red color throughout the paper with the help of our foreign colleagues.

Reviewer 2 Report

The paper is good. The authors could improve it by considering a review of the English Style. Some parts are confusing or without an impact. These are my comments:

  • Abstract: Some lines of this section are difficult to read. The authors could reinforce the gap and contribution of this research.

- Introduction: it is good. Considering the abstract, focus on the gap and contributions. There are some concepts or comments without references. Describe more methodology and possible alternatives. 

- Literature review. It is good and balanced.  

- Basic model: It is good. The author could reinforce the contributions in the final paragraph.

- Model under information Describe more of the steps or research procedure. Explain the procedure more.

- Coordination mechanism based on revenue-sharing and cost-sharing contracts is a good section. 

- Numerical analysis: it is ok. 

- Conclusions: it is the only description. The authors could improve this section by considering aligning this section with the introduction, coordination mechanism, and numerical analysis. 

Author Response

Referees Response Letter

Thank for the comments to improve our paper, and we revised the paper according to the comments as following:

The paper is good. The authors could improve it by considering a review of the English Style. Some parts are confusing or without an impact. These are my comments:

  • Abstract: Some lines of this section are difficult to read. The authors could reinforce the gap and contribution of this research.

- Introduction: it is good. Considering the abstract, focus on the gap and contributions. There are some concepts or comments without references. Describe more methodology and possible alternatives. 

- Literature review. It is good and balanced.  

- Basic model: It is good. The author could reinforce the contributions in the final paragraph.

- Model under information Describe more of the steps or research procedure. Explain the procedure more.

- Coordination mechanism based on revenue-sharing and cost-sharing contracts is a good section. 

- Numerical analysis: it is ok. 

- Conclusions: it is the only description. The authors could improve this section by considering aligning this section with the introduction, coordination mechanism, and numerical analysis. 

Response:

  1. we revised the abstract in red color, and make it easy to understand.
  2. we reinforced the gap in Page 5, in line 211-234, in green color, as below

There were many literatures on the traditional closed-loop supply chain, but few about E-CLSC. There exists significant difference between traditional closed-loop supply chain and E-CLSC, such as operation structure, cooperation mode, recycling process, and so on. Therefore, it is difficult to apply the conclusions of the traditional closed-loop supply chain to E-CLSC directly. With the developing of network and information technology, it is necessary to optimize recycling decision and supply chain coordination specially for E-CLSC in line with the development of e-commerce and big data so as to effectively reduce recycling and remanufacturing cost, improve waste recovery efficiency and economic benefit for enterprise, and thus effectively improve sustainable social development and enhance environmental benefit. Besides, the existing research does not involve the improvement of recycling rate and supply chain coordination at the same time, while improving the recycling rate is an important goal of E-CLSC, and supply chain coordination directly determines the stability of the E-CLSC operation.

Furthermore, there are some literatures referred to the effect of fairness concern or altruistic reciprocity on recycling decision and contract coordination, but there are two problems: firstly, there are too few literatures introducing fairness concern or altruistic reciprocity into E-CLSC, e.g. fairness concern [10,20], altruistic reciprocity [19,35-36], and these literatures do not study the impact of fairness concern and altruistic reciprocity on recycling decision and contract coordination at the same time. Secondly, these literatures assume that the social preference information is symmetric, but the social preference belongs to subjective and private information, and there exits problems of deliberate concealment and disguise. Therefore, it is necessary to investigate the effect of fairness concern and altruistic reciprocity on the recycling decision and contract coordination of E-CLSC simultaneously under symmetric and asymmetric information.

  1. In the conclusion, we add some content to it, and we illustrate our research conclusion, limitation and future direction. The detailed information as below:

There are some limitations as below:

Firstly, in this paper, we only investigate the effect of E-platform’s fairness concern and manufacturer’s altruistic reciprocity on recycling decision optimization and contract coordination in the E-CLSC dominated by manufacturer. For E-CLSC relies on E-platform to recycle waste products and sell remanufactured products, horizontally integrates all production process, recycling process and sale process by network, and achieves the parallel operation of forward logistics, reverse logistics and sales processes so as to improve the efficiency of waste products recycling and remanufacturing obviously. With the development of various intelligent network technologies (e.g. big data, cloud computing, blockchain, etc.), e-commerce will become more popular and thus E-platform will play a more important role in every E-CLSC process including manufacture, remanufacture, recycle and product sale, so E-platform will become a key role in the operation of entire E-CLSC, e.g. JD Mall, Tmall Global, Vipshop, Suning E-shop, etc., and they will dominate E-CLSC with stronger pricing priority ability and strength than small manufacturers, so these large-scale E-platforms usually occupy a dominant position in E-CLSC and E-CLSC is becoming E-platform dominated. It is necessary to consider the E-platform’s altruistic reciprocity and manufacturer’s fairness concern in E-CLSC dominated by E-platform, and investigate the effect of social preference on the decision-making and contract coordination, and check whether the RC contract can coordinate E-CLSC.

Secondly, our model includes a manufacturer, an E-platform and a recycler, and E-platform only cares about the upstream manufacturer’s profit, while ignoring the profit comparison with other competitive E-platforms’ profit. The competitive E-platforms have direct competitiveness and are more likely to care about the profit with each other in the same market position with similar service. Therefore, it is necessary to expand the social preference to E-platforms competing condition in the E-CLSC including multiple E-platforms. Although the complexity and difficulty of calculation will greatly increase, it is closer to the real E-CLSC competition environment. For example, we can set another model including a manufacturer and two competitive E-platforms, and it is necessary to consider profit comparison between E-platform and manufacturer and profit comparison between various E-platforms so as to investigate the effect of social preference on the recycling decision and contract coordination, and check whether flexible contracts (e.g. revenue-sharing and cost-sharing contract, transfer payment contract, two-part pricing contract, etc.) can coordinate E-CLSC and how social preference influence the contract coordination range, so as to propose more practical management strategies for improving recycling efficiency and E-CLSC operation.

Finally, we investigate the impact of E-platform’s fairness concern and manufacture’s altruistic reciprocity on the recycling decision and contract coordination of E-CLSC under symmetry and asymmetry information condition. It is essential to identify the types and intensity of social preference so as to avoid the negative impact of asymmetric social preference on the recycling decision. For example, we can apply the cost-benefit analysis to choose the E-platform to join supply chain only when the revenue is greater than relative cost.

  1. We revised the language problems and grammatical mistakes in red color throughout the paper with the help of our foreign colleagues.

Reviewer 3 Report

1. Line 258 "The manufacturer produces new products and remanufactured products and sell 257 them to E-platform simultaneously at a certain wholesale price".. Should not a Certified Pre-Owned (CPO) be cheaper than a new product ? Please clarify why your assumption is valid

2.  In figure 1a, can a recycler recycles 100% of wastes ? Please clarify why your assumption is valid

3. Please clarify why line 285 is a precondition for a meaningful supply chain system. For subscription based companies, it is conceivable that selling prices can be lower than manufacturing costs. For example, they can run promotion to attract new customers and reduce customer churn rate

Author Response

Referees Response Letter

Thank for the comments to improve our paper, and we revised the paper according to the comments as following:

  1. Line 258 "The manufacturer produces new products and remanufactured products and sell 257 them to E-platform simultaneously at a certain wholesale price".. Should not a Certified Pre-Owned (CPO) be cheaper than a new product ? Please clarify why your assumption is valid.

We add the clarification in Page 6, in line 284-293, in green color, as below:

Although the recycled products cannot be recovered to perfectly new standard through remanufacturing, but it can meet the normal use of consumer after special remanufacture. If we consider the difference between new products and the reprocessed products, we need add another relevant parameter, such as the proximity to the new products or the substitution degree with the new products, which will increase the model complexity seriously, and at the same time, we cannot significantly analyze the impact of social preferences on the recycling decision. Therefore, we assume that the recycled products can have no difference with the new products through remanufacturing, reduce the complexity of the model, and make the effect of social preference on recycling decision more obviously and directly.

  1. In figure 1a, can a recycler recycles 100% of wastes? Please clarify why your assumption is valid.

Similar to question 1, Although the recycled products cannot be recycled at all and only certain percentage of wastes could be recycled in reality. Our research focus on significantly analyzing the impact of social preference on the E-CLSC recycling decision and contract coordination with fewest parameters, so we minimize parameters and avoid unnecessary discussion. But in the future research, we can consider only certain ratio of wastes recycled, and further investigate the effect of social preference on the recycling decision and contract coordination.

  1. Please clarify why line 285 is a precondition for a meaningful supply chain system. For subscription based companies, it is conceivable that selling prices can be lower than manufacturing costs. For example, they can run promotion to attract new customers and reduce customer churn rate

Promotion is a special case, and enterprises usually sell products normally, so the retail price is definitely higher than production cost. In addition, if sales promotion is carried out, the reduction of unit profit and the increase of demand brought by low-price sales promotion should be considered and balanced, so the demand function must be reconstructed, that is, the demand function should include the retail price and the influence coefficient of sale promotion on demand. Obviously, when we consider the effect of promotion on the new products and remanufacture products, the model will be much more complicated, and even the optimal solution cannot be guaranteed. However, the comments put forward by the reviewer are very good. In the future, we can consider the optimal recycling decision and contract coordination of the closed-loop supply chain by considering sale promotion, that is, enterprises sell products at a retail price lower than the cost price, so as to make conclusions adapt to more scenarios.

Round 2

Reviewer 1 Report

I think the authors have already satisfied my concern. Therefore, I recommend acceptation for this paper.

Reviewer 3 Report

Accept in present form